

# Evaluation of the CloudSat surface snowfall product over Antarctica using ground-based precipitation radars

Niels Souverijns[1], Alexandra Gossart[1], Stef Lhermitte[2], Irina V. Gorodetskaya[3], Jacopo Grazioli[4,5], Alexis Berne[4], Claudio Duran-Alarcon[6], Brice Boudevillain[6], Christophe Genthon[6], Claudio Scarchilli[7], and Nicole P. M. van Lipzig[1]

[1]Department of Earth and Environmental Sciences, KU Leuven, Belgium
[2]Department of Geoscience and Remote Sensing, Delft University of Technology, The Netherlands
[3]CESAM - Centre for Environmental and Marine Studies, Department of Physics, University of Aveiro, Portugal
[4]Environmental Remote Sensing Laboratory, Ecole Polytechnique Fédérale de Lausanne, Switzerland
[5]Federal Office of Meteorology and Climatology, MeteoSwiss, Locarno-Monti, Switzerland
[6]Université Grenoble Alpes, CNRS, IRD, Grenoble INP, IGE, France
[7]Laboratory Observation and analyses of Earth and Climate, ENEA, Italy

**Correspondence:** Niels Souverijns (niels.souverijns@kuleuven.be)

**Abstract.** In-situ observations of snowfall over the Antarctic Ice Sheet are scarce. Currently, continent-wide assessments of snowfall are limited to information from the Cloud Profiling Radar on board of CloudSat, which has not been evaluated up to now. In this study, snowfall derived from CloudSat is evaluated using three ground-based vertically profiling 24-GHz precipitation radars (Micro Rain Radars; MRRs). Firstly, using the MRRs long-term measurement records, an assessment of
5 the uncertainty caused by the low temporal sampling rate of CloudSat (one revisit per 2.1 to 4.5 days) is performed. The 10-90[th] percentile temporal sampling uncertainty on the snowfall climatology varies between 30-40 % depending on the latitudinal location and revisit time of CloudSat. Secondly, an evaluation of the snowfall climatology indicates that the CloudSat product, derived at a resolution of 1° latitude by 2° longitude, is able to accurately represent the snowfall climatology at the three MRR sites (biases < 15 %), outperforming ERA-Interim. For coarser and finer resolutions, the performance drops due to
10 higher omission errors by CloudSat. Moreover, the CloudSat product does not perform well in simulating individual snowfall events. Since the difference between the MRRs and the CloudSat climatology are limited and the temporal uncertainty is lower than current CMIP5 snowfall variability, our results imply that the CloudSat product is valuable for climate model evaluation purposes.

## 1 Introduction

The surface mass balance (SMB) of the Antarctic Ice Sheet (AIS) is an important control mechanism in the determination of (future) sea level rise (Gregory and Huybrechts, 2006; Genthon et al., 2009a; Hanna et al., 2013; Ligtenberg et al., 2013; Previdi and Polvani, 2017; Lenaerts et al., 2017). It comprises the sum of snowfall, sublimation / evaporation, melt and blowing snow (van den Broeke et al., 2004). An important component in the SMB of the AIS is snowfall, being the main positive term (Boening et al., 2012). However, snowfall is still poorly constrained in current state-of-the-art climate models and reanalysis



(Genthon et al., 2009b; Bromwich et al., 2011; Palerme et al., 2017; Tang et al., 2018). Tang et al. (2018) indicate a large spread in the annual snowfall amounts of four reanalysis products south of 60° S (differences up to 200 mm year$^{-1}$. Furthermore, they point out that these reanalysis products have low correlation with each other and show contrasting trends in historical snowfall amounts. Models of the Fifth Climate Model Intercomparison Project (CMIP5) simulate historical snowfall rates over the AIS

ranging from 158 to 354 mm year$^{-1}$ (Palerme et al., 2017).

Climate models resolve the different components of the SMB individually. Nevertheless, their evaluation is usually limited to the total SMB (Lenaerts et al., 2012; Wang et al., 2016) as there is a lack of observations of the individual components. For example, snowfall reduced by sublimation is often equated to accumulation records when evaluating climate models. These accumulation records are mainly obtained locally from ice cores and stake measurements (Genthon et al., 2005; Magand et al.,

2007; Eisen et al., 2008; Favier et al., 2013), while continent-wide estimates of the SMB of the AIS are retrieved from satellite data or the integration of distinct observational records (Vaughan et al., 1999; Velicogna and Wahr, 2006; Medley et al., 2014; Hardy et al., 2017). It is not straightforward to relate snowfall rates to accumulation especially at the local scale, as blowing snow often disturbs these records, making the distinction between transported and precipitated snow challenging (Bromwich et al., 2004; Frezzotti et al., 2004; Knuth et al., 2010; Scarchilli et al., 2010; Gorodetskaya et al., 2015; Gossart et al., 2017;

Souverijns et al., 2017a). Moreover, current observed trends in accumulation are increasing faster than predicted by models (Medley et al., 2018). This stresses the need for reliable snowfall observations over the AIS in order to constrain climate models and to get accurate estimates of the future AIS SMB and sea level rise.

In the last decades, several efforts have been made to get accurate estimates of snowfall over the AIS. However, the amount of observations stays limited. In 2010, the first ground-based Micro Rain Radar (MRR) over Antarctica was installed at the Belgian

Princess Elisabeth station (Gorodetskaya et al., 2015). Using disdrometer observations at the surface, a relation between radar reflectivity and snowfall rates was achieved (Souverijns et al., 2017b). In 2015, two more MRRs were installed at Dumont D'Urville station (Grazioli et al., 2017a) and Mario Zucchelli station, for which also reliable snowfall rates were obtained.

Apart from ground-based radar measurements, space-borne observations are also a valuable source of information over the AIS. The Cloud Profiling Radar on board of the CloudSat satellite (Stephens et al., 2002) is the first to provide information

about snowfall on a continental scale over the AIS using the 2C-SNOW-PROFILE product (Wood et al., 2013, 2014). Launched in 2006, it overpasses each location on the AIS within 100 km with a temporal revisit time of seven days or less and has a strong latitudinal dependency (Van Tricht et al., 2016). Palerme et al. (2014) constructed a continental snowfall climatology at a grid of 1° latitude by 2° longitude, including information about the phase and frequency of snowfall. A yearly average snowfall rate of 171 mm year$^{-1}$ over the AIS north of 82° S was found, higher than observations of snow accumulation, but

significantly lower than the CMIP5 ensemble mean (Palerme et al., 2017). Furthermore, the product agrees reasonably well with ERA-Interim reanalysis (Dee et al., 2011) despite the large uncertainties in the retrieval algorithm and the low temporal sampling rate of CloudSat (Palerme et al., 2014).

Although the CloudSat satellite is the first to offer a continent-wide (north of 82°S) estimation of snowfall over the AIS, the evaluation of this product with ground-based observations of snowfall is still limited (Maahn et al., 2014). In this paper, the

CloudSat snowfall product will be compared against observations of the three MRRs that are currently deployed over the AIS.



As a first step, the effect of the low temporal sampling rate of CloudSat on the resulting snowfall climatology is investigated, including an overview of the temporal uncertainty (Sect. 3.1). Next, a climatology is constructed for periods of concurrent observations of the MRRs and CloudSat. The climatology is calculated at different spatial resolutions and evaluated against observations of the three stations. Furthermore, an overview of the discrepancies between CloudSat and the MRR snowfall rates are identified by comparing individual snowfall events (Sect. 3.2). To conclude, a comparison with ERA-Interim reanalysis is performed, currently often used for continent-wide estimates of snowfall over the AIS (Sect. 3.3).

## 2 Material and methods

### 2.1 Ground-based precipitation radars

Local snowfall measurements by precipitation gauges or disdrometers are hindered in polar regions by the high wind speeds concurring with most snowfall events. Therefore, ground-based precipitation radars have been installed at several Antarctic stations, which attain for an independent view on the snowfall component of the SMB over the AIS. At the moment, there are only three locations over the AIS where the instrument is deployed (Fig. 1): (1) the Belgian Princess Elisabeth (PE) station (71°57' S, 23°21' E; 1392 m above sea level), located 173 km from the coast, in Dronning Maud Land, north of the Sør Rondane mountain chain (a detailed description of the setting can be found in Gorodetskaya et al. (2013)). (2) the French Dumont D'Urville (DDU) station (66°40' S, 140°01' E; 41 m above sea level), located at the coast of Terre Adélie (a detailed description can be found in Grazioli et al. (2017a)). (3) the Italian Mario Zucchelli (MZ) station (74°41' S, 164°07' E; 15 m above sea level), located at the coast of Victoria Land in the Terra Nova Bay area, surrounded closely by high mountain chains (a detailed description can be found in Scarchilli et al. (2010)).

The precipitation radars deployed at the three stations (MRRs designed by Metek) are vertically pointing operating at a frequency of 24 GHz (Klugmann et al., 1996; Peters et al., 2002). As these instruments were originally developed to detect liquid precipitation, operational MRR procedures to derive standard radar variables, as e.g. radar reflectivity, were modified for snowfall using the methodology of Maahn and Kollias (2012), increasing the minimum detectable range to -14 dBz in the lowest measurement bins. Radar reflectivity measurements were subsequently converted to snowfall rates using relations specifically developed for the MRR at the PE station (Souverijns et al., 2017b) and at the DDU station (Grazioli et al., 2017a). For the MRR at the MZ station, no relation has yet been developed. As such, the relation obtained at the DDU station was also applied here, as the setting of both stations (located near the coast) can be considered similar.

The MRRs deployed at all stations measure snowfall rate intensity between 300-3000 meters with a vertical height resolution of 100 meters. MRR measurements are available at the minute time scale and are summed to hourly values for most of the applications. It must be noted that the MRR snowfall record is characterised by uncertainties in the radar reflectivity-snowfall rate relation. At the PE station for example, this uncertainty equals ±60 % (Souverijns et al., 2017b). A similar uncertainty range is obtained for the radar-reflectivity snowfall rate relation obtained at the DDU station (Grazioli et al., 2017a).



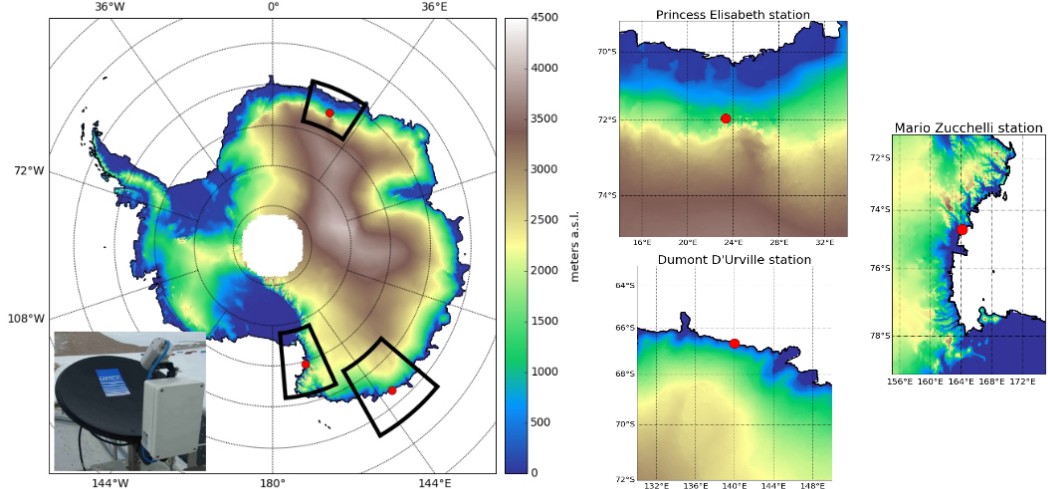

**Figure 1.** Digital Elevation Map of the Antarctic Ice Sheet (Liu et al., 2015) with three insets corresponding to the location of the Micro Rain Radars. Upper: Princess Elisabeth station, right: Mario Zucchelli station, lower: Dumont D'Urville station. The inset at the bottom left shows the Micro Rain Radar at the Princess Elisabeth station.

## 2.2 CloudSat snowfall climatology

Apart from ground-based radars, the Cloud Profiling Radar on board of the CloudSat satellite, nadir-looking and operating at 94 GHz, has been used to derive snowfall rate estimates over the AIS (Stephens et al., 2002). The 2C-SNOW-PROFILE product (Wood et al., 2013, 2014) derives snowfall rates from radar reflectivity measurements. The relation between radar reflectivity

and snowfall rates is derived using a priori estimates of snow particle size distribution, microphysical and scattering properties (Wood et al., 2013, 2014). The comparison between the ground-based and space-borne radars is facilitated as snowfall rates are derived using similar procedures by the MRRs (Souverijns et al., 2017b). Furthermore, as the optimal estimation retrieval (Rodgers, 2000) is used to derive the 2C-SNOW-PROFILE product, the relation between radar reflectivity and snowfall rates is variable over the AIS. This is considered important as this relation varies significantly from coastal to inland regions (Souverijns

et al., 2017b).

  The Cloud Profiling Radar of CloudSat has a narrow swath-width (1.7 km by 1.3 km footprint) and provides snowfall rate profiles divided into 150 vertical bins at a resolution of 240 m. In order to remove the effects of ground clutter, the bin closest to the surface that is useful is located at 1200 m above ground level. From this data, a snowfall climatology map was created by Palerme et al. (2014) for the AIS by mapping the 2C-SNOW-PROFILE tracks over a grid of 1° latitude by 2° longitude.

For each orbit, one snowfall rate value per grid cell that is overpasses by CloudSat is retained, taken as the mean value of all snowfall rates in this grid cell. At a spatial resolution of 1° latitude by 2° longitude, the temporal revisit time of CloudSat for each grid cell is five days at maximum (Palerme et al., 2014).



## 2.3 Comparative analysis

CloudSat provides currently the only continent-wide snowfall product over the AIS. As no ground-based precipitation estimates have been available up to now, this product has not been evaluated yet. CloudSat has been operational since 2006. However, due to battery issues, it is not longer able to operate during the night orbit (i.e. at the non-sunlit side of the earth). As such, no

snowfall rate measurements are obtained during austral winter season since 2011. The MRR at the PE station was installed in January 2010 and was planned to operate continuously throughout the year. However, due to power cuts at the station, austral winter observations are only available in 2012, limiting the collocated data coverage to the periods of the austral summer (Fig. 2). Next to this, no field campaign took place during the 2016-2017 austral summer, leaving a data gap of 18 months since May 2016. In total, 928 days of collocated measurements of both CloudSat and the MRR are available at the PE station.

The MRR at the DDU station was installed in December 2015, operating nearly continuously until present time, leading to 519 days of collocated measurements (Fig. 2). At the MZ station, the MRR is operating continuously since November 2016, after one summer season of measurements in 2015-2016, accounting for 333 days of collocated measurements (Fig. 2). As no full year of collocated measurements between CloudSat and the MRRs is available, the comparative analysis will be limited to the austral summer periods (denoted in purple in Fig. 2). Since our main interest lies in the measurement of snowfall at

the surface, the lowest usable measurement bin of both instruments is considered in the analysis. The data acquisition height difference between CloudSat (1200 m a.g.l.) and the MRRs (300 m a.g.l.) accounts for a typical underestimation of 9-11 % in total snowfall amount by CloudSat compared to the MRR at the PE station (Maahn et al., 2014), while at DDU station this equals 13 %, caused by sublimation in these low layers of the atmosphere (Grazioli et al., 2017b). Furthermore, sublimation persists towards the surface, also influencing the layer between the lowest measurement bin of the MRR (i.e. 300 m a.g.l.) and

the surface, where typically an inversion is present (Grazioli et al., 2017b; Souverijns et al., 2017b). The discrepancy in the lowest 300 m of the atmosphere is not considered in this study.

     Four experiments will be described in this paper. As a first step, a statistical analysis is executed in order to obtain an overview of the uncertainty caused by the low temporal revisit time of CloudSat (Palerme et al., 2014; Van Tricht et al., 2016). The revisit time of CloudSat equals several days for most of the locations on the AIS. In case a spatial resolution of 1° latitude

by 2° longitude is chosen (conform Palerme et al. (2014)), the revisit time is on average 4.7 days for the DDU station, 2.5 days for the PE station and 2.1 days for the MZ station. The MRRs achieve snowfall rate estimates on a one-minute temporal resolution. By subsampling from the MRR record during periods of collocated MRR and CloudSat measurements, a similar temporal sampling resolution as CloudSat is obtained, which can be compared to the full MRR record. The systematic sampling technique is applied to the MRR snowfall record (randomly selecting the starting point, while using a fixed periodic interval

for subsequent observations; 10.000 bootstraps).

     Next, the total snowfall amounts obtained by CloudSat and the MRRs are calculated for all periods with collocated measurements (Fig. 2). The methodology of Palerme et al. (2014) is used to obtain snowfall rate estimates of CloudSat. The AIS is overlaid by a grid. Each time the CloudSat satellite overpasses a grid cell, one sample is retained by taking the average of all observations within the grid cell. The spatial resolution is fixed in Palerme et al. (2014) at 1° latitude by 2° longitude. However,



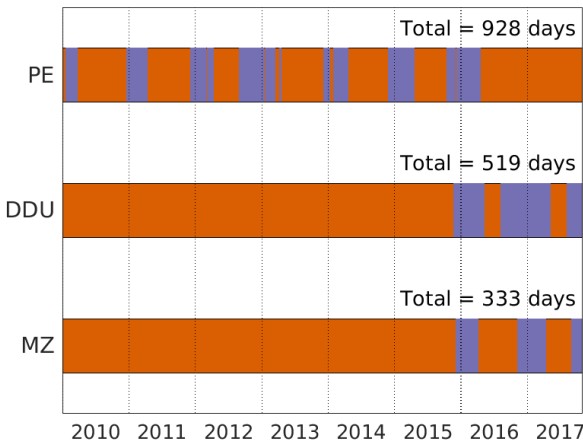

**Figure 2.** Overview of periods with concurrent CloudSat and MRR measurements for the Princess Elisabeth (PE), Dumont D'Urville (DDU) and Mario Zucchelli (MZ) station denoted in purple. Vertical dotted lines denote the start of a year.

by varying the spatial resolution of the grid overlaying the AIS (and therefore the distance between the satellite overpasses that are taken into account and the MRRs), a different performance is expected. As such, the analysis is performed for several spatial resolutions varying from 0.1° latitude by 0.2° longitude to 2° latitude by 4° longitude in steps of 0.1° latitude by 0.2° longitude.

Apart from the total snowfall amount, individual snowfall events recorded by both the MRRs and CloudSat are investigated. Individual CloudSat overpasses in the grid box over the station are averaged and compared to measurements of the MRRs. This analysis is executed using different spatial resolutions (varying from 0.1° latitude by 0.2° longitude to 2° latitude by 4° longitude) in order to investigate the effect of the spatial resolution on the match in snowfall amounts.

   Lastly, a comparison between the MRRs, CloudSat and ERA-Interim reanalysis (Dee et al., 2011) is executed for the three
stations. ERA-Interim reanalysis data is generally considered one of the best reanalysis products regarding snowfall over Antarctica, however still very biased (Bromwich et al., 2011; Medley et al., 2013). Notwithstanding the availability of CloudSat and MRR snowfall records, their measurements are not yet assimilated in ERA-Interim reanalysis. As such, all products are independent. Total snowfall amount estimates over the full measurement period will be compared. Furthermore, the performance of individual event detection of CloudSat and ERA-Interim reanalysis is investigated.

## 3   Results and Discussion

### 3.1   Temporal sampling frequency of CloudSat

Considering the full MRR snowfall record, the precipitation climate over Antarctica is characterised by a limited number of events attaining for large snowfall amounts (Fig. 3 & 4), mainly driven by large-scale circulation (i.e. cyclonic activity in



the circumpolar trough; (Gorodetskaya et al., 2013, 2014; Souverijns et al., 2017a)). The snowfall rate distribution is highly skewed to the right and most stations are not characterised by a clear seasonality in snowfall amounts (Fig. 4). It is noted that precipitation observations in winter are scarce (Sect. 2.3), while interannual precipitation variability can be large. At the PE and MZ station, snowfall events of highest intensities are limited to the austral spring (SON) and summer season, while

during austral winter, less large snowfall events are recorded. This complies with van Lipzig et al. (2002) in their study of the seasonality of the SMB over Dronning Maud Land. For the DDU station, more high-intensity snowfall events are observed. Seasonally, the lowest snowfall amounts are obtained during austral summer, while highest contributions to the total snowfall record are obtained during the other months, confirming the results of Grazioli et al. (2017a).

Snowfall events over Antarctica (with total precipitation amount of 1 mm w.e. during the course of the event) generally

span multiple hours (15 hours on average for the PE station (Souverijns et al., 2017a)). This is much shorter than the interval between two overpasses of CloudSat using the resolution of Palerme et al. (2014). This revisit time equals on average 2.5 days for the PE station, 4.7 days for the DDU station and 2.1 days for the MZ station, which is fully determined by their latitudinal location. Therefore, snowfall events are often missed (several examples are visible in Fig. 3). In addition, there is a strong variability in snowfall rates throughout individual events (see e.g. Fig. 3). One overpass every couple of days is therefore not

representative for individual snow storm variability.

In order to get an estimate of the uncertainty induced by the low temporal sampling frequency of CloudSat, systematic sampling is applied on the MRR snowfall record (available on the minute time-scale). For the MZ station for example, the revisit time equals approximately 2.1 days. As such, subsamples are extracted from the MRR record with an interval of 2.1 days. In order to get a fair estimate of the temporal uncertainty induced by the CloudSat temporal revisit time, each of the MRR

subsamples needs to cover a time period. CloudSat has a narrow swath width. During a CloudSat overpass close by the station, a spatial area within the grid box of 1° latitude by 2° longitude is covered by its track (see Sect. 2.2). The distance of this track within the grid box is converted to a time period, i.e. if the track is 130 km long within the grid box and the wind speed at 300 m a.g.l. (which is obtained from ERA-Interim reanalysis data over the stations (Dee et al., 2011)) equals 20 km h$^{-1}$, the MRR subsample covers a time period of 6.5 hours. On average, this time period equals 7.2, 7.4 and 6.9 hours respectively for

the PE, DDU and MZ station. As such, in order to get a correct estimation of the CloudSat temporal uncertainty, in case of the example for the MZ station, for each bootstrap a subsample of 6.91 hours is extracted every 2.1 days (Fig. 5).

For all stations, generally an increase in the uncertainty of the total snowfall amount is observed when decreasing the temporal sampling frequency of data acquisition (Fig. 5). In case less data is available, more uncertain estimates of the total snowfall amount are obtained. For the CloudSat temporal revisit time of Palerme et al. (2014) (2.5 days for the PE, 4.7 days for

the DDU and 2.1 days for the MZ station) large uncertainties on the total snowfall amounts are obtained. The 10-90[th] percentile uncertainty equals [-31 % +10 %] for the PE station, [-37 % +45 %] for the DDU station and [-55 % +36 %] for the MZ station (Fig. 5). Highest uncertainties are found for the DDU and MZ station. For the DDU station, this can be attributed to the low revisit time of CloudSat. Generally, an increase in uncertainty is observed when lowering the revisit time (Fig. 5). For the MZ station, this might be attributed to the short time period of concurrent observations or the highly variable topography of the area

surrounding the station (Fig. 1). As such, depending on the location on the ice sheet and revisit time of CloudSat, the temporal





(a) Princess Elisabeth station

(b) Dumont D'Urville station

(c) Mario Zucchelli station

**Figure 3.** Snowfall rates (mm w.e. h$^{-1}$) during March 2016 at the three stations derived from the MRRs (blue bars), the grid box comprising each of the three stations in ERA-Interim reanalysis (green) and the average of the CloudSat overpasses in the grid box (1° latitude by 2° longitude) comprising each of the three stations following the approach of Palerme et al. (2014) (red).





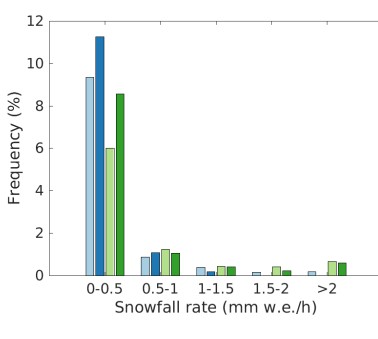
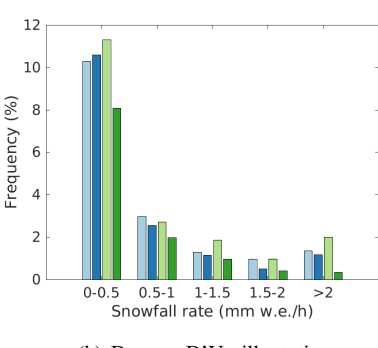
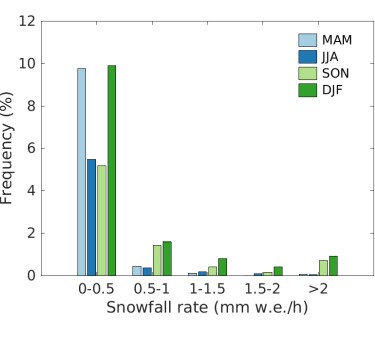

|(a) Princess Elisabeth station|(b) Dumont D'Urville station|(c) Mario Zucchelli station|

**Figure 4.** Seasonal variability of snowfall amounts derived from the MRRs at the three stations.

uncertainty varies between 30-40 % with lower values for regions towards the southern part of the ice sheet. This uncertainty is lower than current CMIP5 model variability (Palerme et al., 2017), showing the potential of CloudSat for evaluation purposes. Apart from the uncertainty induced by temporal sampling, the CloudSat snowfall product is characterised by high uncertainties (between 1.5 and 2.5 times the snowfall rate (Palerme et al., 2014)). As such, interpretations should still be done with care.

Apart from considering the uncertainty on the total snowfall amount, also a median total snowfall amount is achieved from the bootstrapping simulations (Fig. 5). Considering the CloudSat temporal resolution, on average the median total snowfall varies compared to the full MRR snowfall record. For the PE station, an overestimation of 4 % was found, while at DDU and MZ station, a bias of respectively -2 % and +10 % is observed. These biases can be attributed to the skewed distribution of precipitation at the stations (Fig. 7) and needs to be considered when using the CloudSat climatology for model evaluation of

snowfall rates over Antarctica, together with the underestimation due to sublimation (Sect. 2.3).

Regarding extreme snowfall rates, very high uncertainties are found for typical CloudSat temporal sampling frequencies (Fig. 5) and equals [-21 % +72 %], [-38 % +52 %] and [-43% +108 %] for respectively the PE, DDU and MZ station. Furthermore, also a high variability in the median 90[th] percentile snowfall rate of all bootstrapping simulations compared to the value obtained for the full snowfall record is observed.

**3.2   CloudSat total snowfall amount and error identification**

Long-term ground-based snowfall measurements during which concurrent measurements with CloudSat were made, are available for seven austral summer seasons at the PE station, attaining for 928 days. During this time period a total number of 952 mm w.e. of snowfall was registered, approximately 1.03 mm w.e. day$^{-1}$. At the DDU station, concurrent snowfall rate estimates are available for 519 days (three austral summer seasons). A total snowfall amount of 1113 mm w.e. was attained,

leading to average snowfall amounts of 2.14 mm w.e. day$^{-1}$. At the MZ station, during 333 days, a total of 608 mm w.e. was measured (i.e. 1.83 mm w.e. day$^{-1}$). It should be noted that at the MZ station, snowfall events are often of local origin induced by a mixing of warm coastal air from Terra Nova Bay with cold katabatic winds from the mountains (Carrasco et al., 2003; Sinclair et al., 2010). The average daily snowfall amount at the DDU and MZ station is approximately double the amount at







**Figure 5.** Boxplots showing the uncertainty when applying systematic sampling on the MRR snowfall record (10.000 bootstraps) using different temporal sampling frequencies (x-axis, D denotes days). Total snowfall amounts during collocated periods of MRR and CloudSat measurements (left) and the 95[th] percentile snowfall rate (right) are shown. The bottom and top edges of the boxplot indicate the 25-75[th] percentile (dark pink shading), while the whiskers denote the 10-90[th] percentile (light pink shading). The red line denotes the median.



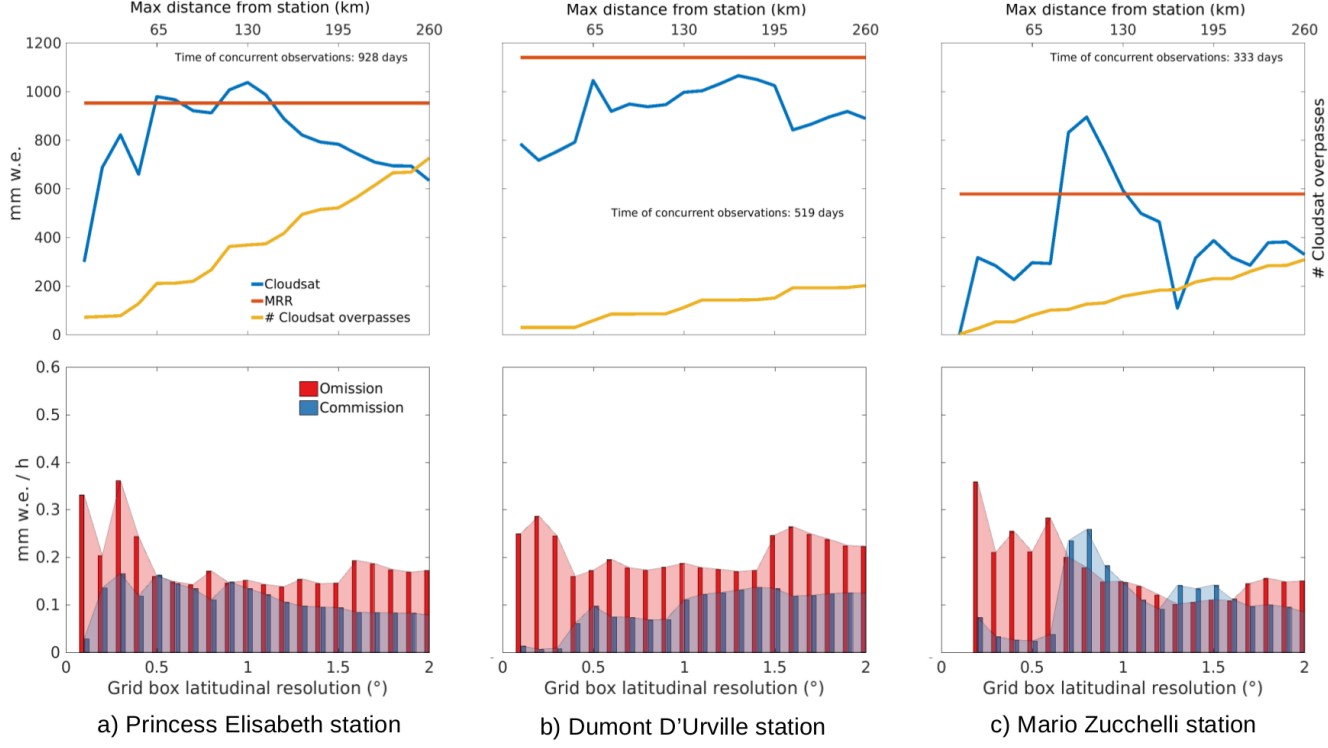

a) Princess Elisabeth station   b) Dumont D'Urville station   c) Mario Zucchelli station

**Figure 6.** (first row) Overview of the total snowfall amounts for the three stations as observed by CloudSat and the Micro Rain Radars during the periods of collocated measurements (Fig. 2). (second row) Individual snowfall event error analysis. MRR snowfall rates are considered truth. The x-axis denotes different spatial resolutions of the CloudSat climatology (grid box longitudinal resolution = 2 * grid box latitudinal resolution).

the PE station. Those two stations are located at the coast of the AIS near sea-level, while the PE station is located 173 km inland at the edge of the Antarctic plateau (Fig. 1). Most of the snowfall originating from cyclone activity in the circumpolar trough has already been deposited upstream of the station due to orographic rising of the air masses (Souverijns et al., 2017a).

Depending on the maximal distance between the CloudSat overpasses and the stations (i.e. the spatial resolution of the grid covering the AIS), a different number of CloudSat overpasses is available for the construction of the total snowfall amount for each grid cell (see Sect. 2.2). For the PE station, in case we only take CloudSat overpasses close to the station into account, i.e. for example a spatial resolution of 0.3° latitude by 0.6° longitude (overpasses within approximately 40 km of the station), only 77 overpasses are available for the calculation of the total snowfall amount in the grid box over the PE station, leading to a temporal revisit time of approximately 12 days (Fig. 6). In case we increase the CloudSat spatial resolution to 2° latitude and 4° longitude (overpasses within approximately 250 km of the station) 726 samples are available, i.e. one sample every 1.3 days.





Apart from comparing the total snowfall amount detected by both the MRR and CloudSat, individual snowfall events detected by both instruments are investigated. Assuming the MRRs define the ground truth, for each snowfall event detected by both instruments, the average omission (misses by CloudSat) and commission errors (overestimations by CloudSat) are calculated (Fig. 6). In order to facilitate the comparison, MRR snowfall rates are calculated by averaging snowfall rates over a time

period following the same procedure as in Sect. 3.1. This time period depends on the the spatial resolution of the grid and the wind speed at 300 m a.g.l.. For example, if the grid has a spatial resolution of 1° latitude by 2° longitude (i.e. with a maximal distance of 130 km between the edges of the grid box) and the wind speed equals 20 km h$^{-1}$, the MRR record is averaged over 6.5 hours. The minimal MRR averaging period is one hour). Using this methodology, one has to assume that the precipitation systems are stationary, which is not valid over highly variable topography (see Sect. 2.3). This source of error needs to be taken

into account when comparing both instruments.

For coarse spatial resolutions, CloudSat underestimates the total snowfall amount compared to the MRR records for each of the three stations (Fig. 6). For these larger spatial scales, CloudSat overpasses are averaged over longer distances. As snowfall amounts are non-stationary, erroneous estimates can be obtained, leading to both omission and commission errors on both the individual event scale as the statistics (Fig. 6). Furthermore, more CloudSat samples are available at higher latitudes

(Palerme et al., 2014). As snowfall rates decrease with latitude (and altitude), which is valid for the PE and DDU station, an underestimation of the snowfall amount (high omission errors) at all stations is observed at coarse spatial resolutions (Fig. 6).

This indicates that fine spatial resolutions are preferred in order to obtain more reliable matches between individual events of CloudSat and the MRRs. However, for the finest spatial resolutions, also large omission errors are identified (Fig. 6). Despite the higher accuracy of MRR measurements and CloudSat overpasses that are closer to the stations, the amount of overpasses is too

low to capture enough high-intensity snowfall events (Fig. 6). As the distribution of snowfall amounts is skewed towards high precipitation numbers (Fig. 4), high-intensity snowfall events are missed leading to an underestimation of the total snowfall amount, which is indeed observed for all stations (Fig. 6).

For intermediate spatial resolutions, reasonable agreements between CloudSat and the MRRs are obtained (Fig. 6). At the PE station, an almost perfect match between snowfall estimates is found for spatial resolutions between 0.5° latitude by 1°

longitude and 1.2° latitude by 2.4° longitude (differences <10 %). For the DDU station, the underestimation of snowfall amounts by CloudSat is limited to 15 % between 0.5° latitude by 1° longitude - 1.5° latitude by 3° longitude. These biases fall within the error margins of the temporal sampling uncertainty (Sect. 3.1). The wider range of accurate snowfall estimates for the DDU station can be attributed to their topographic location. The station is located at the coast of the AIS in a smoothly changing topographical area, minimising snowfall differences (Fig. 1). For the PE station, coarser spatial resolutions imply

snowfall rates from the Antarctic plateau and the coast to be taken into account. Furthermore, the PE station is located near the edge of the Sør Rondane mountain ridge, a highly variable terrain regarding topographic height differences (Fig. 1), leading to a high variability in snowfall rates (Souverijns et al., 2017a). For the MZ station, larger differences between the MRR and CloudSat snowfall amount estimates are obtained. This can be attributed to three factors. First, the station is located close to a large mountain ridge, characterised by highly variable snowfall amounts depending on height, which is difficult to capture

adequately by a CloudSat single track. Second, mesoscale snowfall events develop at the station through the interaction of





warm ocean and cold katabatic air (Carrasco et al., 2003; Sinclair et al., 2010). These mesoscale events are easily missed by CloudSat. Third, concurrent measurements are only available for 333 days. As such, the sample of CloudSat observations is small. This attributes for example for the large jump in snowfall amounts which is observed when increasing the grid box resolution from 0.6° latitude by 1.2° longitude to 0.7° latitude by 1.4° longitude (Fig. 6). This step attributed for the addition of

two major snowfall events, doubling the total snowfall amount that was detected before within the range of 0.6° latitude by 1.2° longitude. In order to erase the influence of single snowfall events, a long-term record of snowfall amounts is indispensable. In Palerme et al. (2014) a grid box width of 1° latitude by 2 ° longitude is used, leading to an accurate estimation of the total snowfall amount based on the analysis above for all three stations. However, for locations close to highly variable topography, erroneous estimates might still be obtained.

For intermediate spatial resolutions, lowest omission errors are observed for all three stations (Fig. 6). However, here, commission errors are generally higher compared to coarse or fine spatial resolutions. The main difference between intermediate and coarse / fine spatial resolutions is that omission errors approximately equal commission errors. As such, the amount of snowfall that is missed by CloudSat approximately equals the amount of false positive snowfall detections. Consequently, when taking long-term averages of CloudSat snowfall rates, an accurate estimate of the total snowfall amount compared to

the MRRs is obtained (Fig. 6). One must understand that the accurate total snowfall amounts obtained by CloudSat can not be attributed to the fact that the satellite is recording correct individual snowfall quantities, but to the fact that omission and commission errors cancel each other out. Consequently, it can be concluded that CloudSat is not the right tool to investigate individual snowfall events / synoptic events at a single location.

As such, the spatial resolution of Palerme et al. (2014) (1° latitude by 2° longitude) gives an accurate representation of the

total snowfall amount for the three stations. In case the distribution of snowfall amounts registered by the MRRs and CloudSat is analysed for this spatial resolution, a clear underestimation of extreme snowfall rates by CloudSat is observed for all three locations in both the distribution and when directly comparing individual events (Fig. 7). As stated above, the underestimation of (the frequency of) large events is the main reason for omission errors (Fig. 6). Furthermore, for all stations, CloudSat is found to detect a higher frequency of snowfall events (Fig. 7). These events often attain for low snowfall rates and are not

detected by the MRRs. As the CloudSat domain spans several tens of kilometers at a resolution of 1° latitude by 2° longitude, it often detects small snowfall events near the station. The detection of these small-scale snowfall events is the main contributor to commission errors compared to the MRRs at this spatial resolution (Fig. 6). In addition, the direct comparison between individual events detected by the MRRs and CloudSat shows a large spread and low correlation (Fig. 7). This indicates again that CloudSat is not able to capture individual snowfall events adequately at a single location.

## 3.3 Comparison with ERA-Interim reanalysis

The total snowfall amount estimate of CloudSat using the spatial resolution of Palerme et al. (2014) showed reasonable agreement with MRR total snowfall amounts (Fig. 6). Apart from CloudSat, no integrated snowfall product is available over the AIS (north of 82°S), apart from accumulation records, climate model simulations and reanalysis. ERA-Interim reanalysis is often taken as a reference regarding the Antarctic-wide snowfall product, however still strongly biased (Bromwich et al., 2011). An



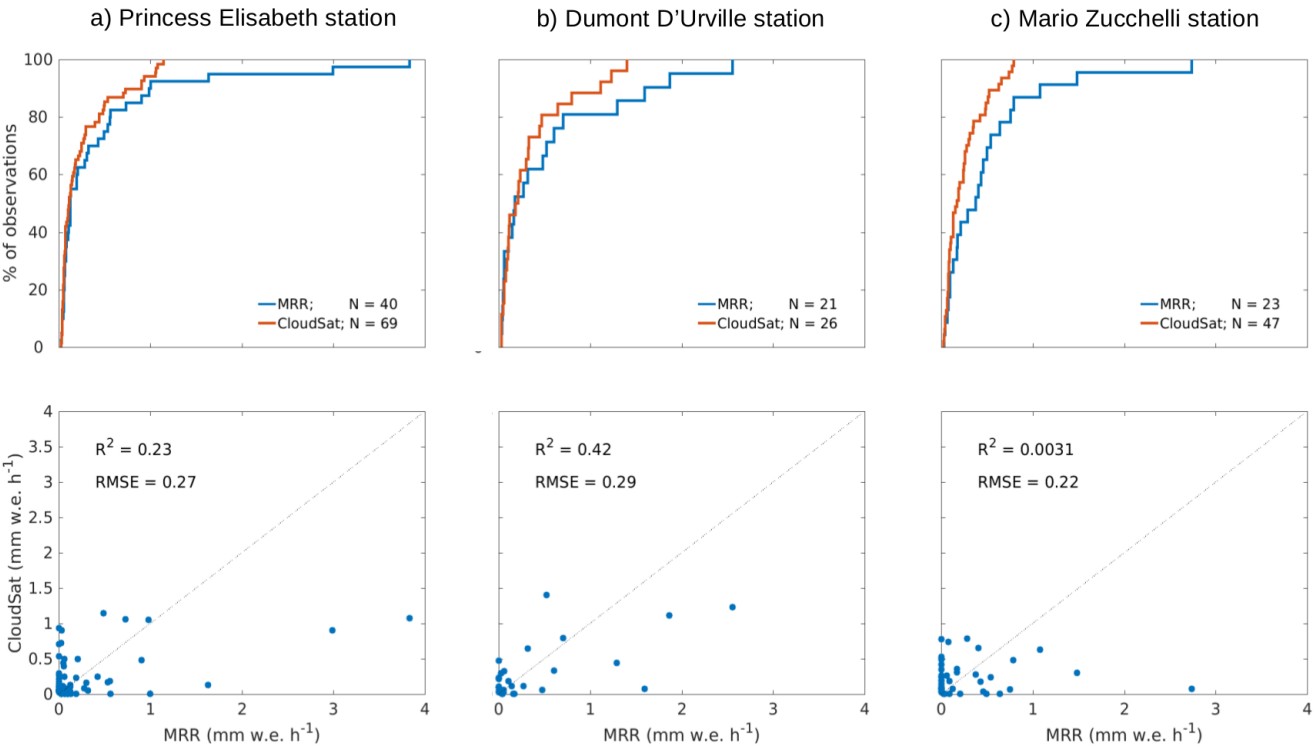

**Figure 7.** (first row) Empirical cumulative distribution of MRR and CloudSat snowfall events at a spatial resolution of $1°$ latitude by $2°$ longitude. (second row) Direct comparison between MRR and CloudSat individual snowfall events. $R^2$ denotes the adjusted coefficient of determination, RMSE is the root mean square error, while the thin line is the bisector.

assessment of the accuracy of the CloudSat snowfall product compared to ERA-Interim reanalysis is therefore viable. For the period of concurrent measurements of MRRs and CloudSat, ERA-Interim reanalysis snowfall amounts are extracted and daily average snowfall amounts are calculated (Table 1). As was shown in Sect. 3.2, a reasonable agreement is observed between CloudSat and MRR average snowfall amounts for all stations. Regarding ERA-Interim reanalysis, for both the PE and MZ

5  station, the daily average snowfall amount is underestimated (respectively by 18 % and 45 %), while for the DDU station, ERA-Interim reanalysis outperforms the CloudSat snowfall estimate (bias is limited to 6 %). At the DDU station, daily radiosoundings are executed which are assimilated in ERA-Interim reanalysis, adding to the performance of this product over the station, explaining its good performance compared to the MRR, even for a derived variable like snowfall. During austral summer, a similar assimilation is conducted at the MZ station. However, here, the performance of ERA-Interim reanalysis

10  snowfall here is still deficient.

Apart from the long-term evaluation, also individual snowfall events can be investigated. For ERA-Interim reanalysis data, daily snowfall amounts are compared with MRR records (Fig. 8). For a fair comparison with CloudSat, the same time frame is chosen (Fig. 2). ERA-Interim reanalysis generally achieves better results when simulating individual snowfall events including



| Station | MRR | CloudSat | ERA-Interim reanalysis |
|---|---|---|---|
| Princess Elisabeth | 1.03 | 1.12 | 0.84 |
| Dumont D'Urville | 2.14 | 1.92 | 2.26 |
| Mario Zucchelli | 1.83 | 1.78 | 1.01 |

**Table 1.** Daily average snowfall amounts (mm w.e. day$^{-1}$) for the concurrent periods displayed in Fig. 2. CloudSat snowfall amounts are derived for the grid specified by Palerme et al. (2014).

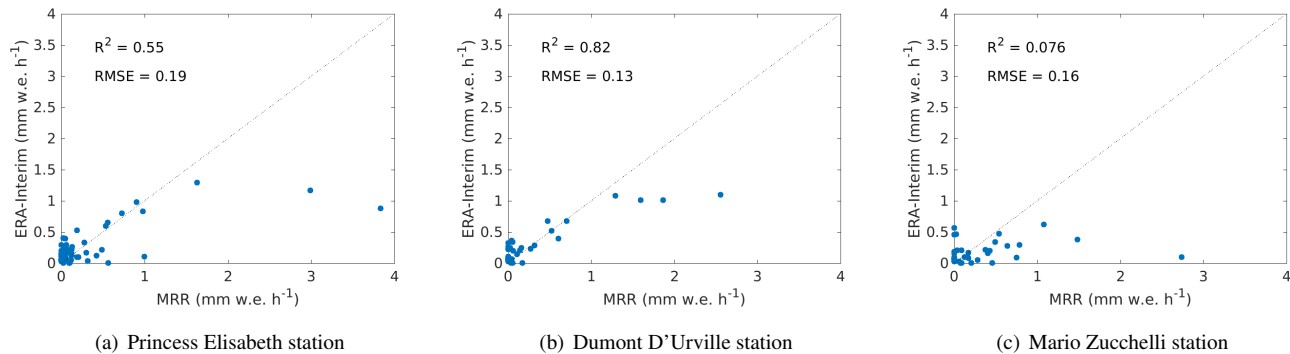

(a) Princess Elisabeth station     (b) Dumont D'Urville station     (c) Mario Zucchelli station

**Figure 8.** Daily snowfall amount comparison between ERA-Interim reanalysis and the MRR. $R^2$ denotes the adjusted coefficient of determination, RMSE is the root mean square error, while the thin line is the bisector.

higher correlations compared to the performance of CloudSat (Fig. 7). For both the PE and MZ station, ERA-Interim reanalysis underestimates the snowfall amount of large events, attaining for omission errors similar to CloudSat (see Sect. 3.2). However, smaller snowfall events are much better captured compared to CloudSat (see also Fig. 3). For the DDU station, which is located close to the coast, a good agreement for large snowfall events is observed, confirming the results deduced from table 1.

5 However, for this station, commission errors are higher. For the validation and identification of individual snowfall events, the ERA-Interim reanalysis product therefore outperforms the CloudSat-derived product.

## 4 Conclusions

The Cloud Profiling Radar on board of the CloudSat satellite is the only instrument from which snowfall rates can be derived over the whole Antarctic Ice Sheet (AIS) at current times (north of 82°S). However, up to now the product has not been

10 evaluated with ground-based observations. In 2010, a Micro Rain Radar (MRR) was installed at the Princess Elisabeth (PE) station in Dronning Maud Land at a distance of 173 km from the coast. In 2015, two more MRRs were set up at the Dumont D'Urville (DDU) and Mario Zucchelli (MZ) station in respectively Terre Adélie and Terra Nova Bay, both located at coastal areas. This paper presents a comparison between these MRRs and CloudSat for periods of concurrent measurements, which is mainly restricted to austral summer periods.



The CloudSat satellite has a temporal revisit time of several days over most of the AIS. Using systematic sampling on the full MRR record and a bootstrapping methodology, it was found that the 10-90[th] percentile uncertainty on total snowfall amounts varies around 30-40 % depending on the latitudinal location of the station. The uncertainty is lower compared to state-of-the-art CMIP5 models, showing the potential of evaluating climate models with this climatology. However, the CloudSat snowfall product is also characterised by high uncertainties due to the relation between radar reflectivity and snowfall rates, which should also be taken into account in the interpretation of this snowfall product. The low temporal sampling frequency does not only impact the uncertainty, but also the median snowfall amount estimate. A variability in the total snowfall amount compared to a continuous record of up to 10 % was observed depending on the station.

The CloudSat total snowfall climatology is highly dependent on the resolution of the grid depending on the spatial resolution of the grid. Choosing a coarse spatial resolution increases the number of samples per grid box, but leads to the inclusion of information from larger distances. Furthermore, in case of coarse spatial resolutions, snowfall amounts are smoothed out, more southern precipitation is included and an underestimation of the total snowfall amount is obtained. In case a fine spatial resolution is preferred, more accurate estimations are obtained. However, the amount of CloudSat samples is low. As such, distinct snowfall events are missed, leading again to an underestimation of the total snowfall amount. The best total snowfall amount estimate compared to the MRR records is obtained for spatial resolutions close to 1° latitude by 2° longitude, which equals to the spatial resolution chosen by Palerme et al. (2014) to obtain their snowfall climatology map for the AIS. However, the good agreement between the MRRs and CloudSat regarding total snowfall amounts can not be attributed to accurate snowfall rate recordings of CloudSat on an event basis, but rather to the fact that omission errors are compensated equally by commission errors for this spatial resolution.

The CloudSat snowfall climatology provides very good results compared to MRR total snowfall amount records for all three stations, showing the skill of CloudSat for the estimation of the snowfall climatology over the AIS, outperforming ERA-Interim reanalysis. ERA-Interim reanalysis total snowfall records generally underestimate the MRR snowfall amounts at the PE and MZ station. At the DDU station, a better performance is achieved, which is mainly related to the assimilation in ERA-Interim reanalysis of a daily radiosounding collected at the DDU station. Nevertheless, the assimilation of radiosoundings does not ameliorate the performance of ERA-Interim reanalysis at the MZ station. However, for individual snowfall event identification, ERA-Interim reanalysis outperforms CloudSat for all stations.

CloudSat's primary skill is the estimation of the snowfall climatology, offering adequate estimations compared to MRR records and outperforming ERA-Interim reanalysis approximations. However, the CloudSat snowfall climatology is characterised by large uncertainties inherent to the product and the temporal sampling frequency. Apart from that, CloudSat is not advised for the validation of individual snowfall events. For this, ERA-Interim reanalysis achieves better skill. In order to increase confidence in the CloudSat snowfall product at the local scale, more ground-based measurements, including scanning radars, are necessary. Furthermore, with the future launch of the EarthCare satellite, year-round estimates of precipitation will become available again for the AIS, attributing to better precipitation estimates over the continent.



*Data availability.* CloudSat data is freely available via the CloudSat Data Processing Center (http://www.cloudsat.cira.colostate.edu/). Data from the Micro Rain Radar at the Princess Elisabeth station can be obtained from the database on http://www.aerocloud.be. Data from the Micro Rain Radar at Dumont D'Urville station are available at https://doi.pangaea.de/10.1594/PANGAEA.882565, while for the Micro Rain Radar at Mario Zucchelli station, data will be made publicly available as soon as possible (contact person: claudio.scarchilli@enea.it).

5  *Competing interests.* The authors declare that they have no conflict of interest.

*Acknowledgements.* This work was supported by the Belgian Science Policy Office (BELSPO; grant number BR/143/A2/AEROCLOUD) and the Research Foundation Flanders (FWO; grant number G0C2215N).





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
