# Peer review of "Evaluation of the CloudSat surface snowfall product over Antarctica using ground-based precipitation radars"

_The Cryosphere, 2018_

## Referee Comment (RC1) · Anonymous Referee #1 · 17 Jul 2018

This study evaluates the surface snowfall measured derived by CloudSat by comparing it with the product from ERA and Micro Rain Radar signals over Antarctica. The study brings valuable information to the scientific community to better understand changes in snow cover over Antarctica. It uses a combination of ground instruments, satellite and reanalysis data. It was found that CloudSat does not measure well snowfall because of the lower time resolution of passes of the regional whereas ERA covers the entire continent with 3 hourly outputs. The snowfall measured by the MRR is similar to ERA because it can capture single events as opposed to CloudSat. This study contributes to better understand the evolution of snow cover in Antarctica. Finally, the English and the clarity of some figures should also be improved.

[Figure]

These are the specific comments/questions:

1. P.3, line 27-31: It would be useful to explain in an Appendix the choice of the Z-S relationship chosen and how it was compared to other available data used to measure snowfall at each site (other than CloudSat). Also, could you clarify in the manuscript how the 3 sets of data were compared? If I understood correctly they don't provide the snowfall information at the same altitude. For example, it is indicated in the manuscript that the lowest level where the snowfall is estimated by CloudSat is at 1200 m and the MRRs is at 300 m above ground level. I assumed that ERA snowfall is probably produced at the surface. How these differences affect the results of your study?

2. P.4, Figure 1: Could you add the name of the stations on the map with the acronym used in the text and other figures?

3. P.5, line 25, Please double check the guidelines for references. The reference should be (conforme Palerme et al., 2014) instead of (conforme Palerme et al. (2014)).

4. P.6, line 5, How is a snowfall event defined? Were they defined per day or per time period when snow accumulated at the ground (or at the lowest MRR level)?

5. P.7, line 16-26, there are 3 times "in order to get" in the same paragraph. It should be reworded.

6. P.8, Figure 3, It is surprising to see that the MRR misses many events detected by ERA. It should be further discussed in the manuscript. Also, a legend should be added to the figure.

7. P.9, figure 4, Is it possible to do the same figure from ERA and maybe CloudSat? It would be interesting to see how well they compare among each other if possible.

8. P.9, line 9, Should it be Fig 4 instead of Fig 7?

9. P.10, Figure 5: Add the title of the y-axis on the left column.

10. P.11, Figure 6: Add the title of the y-axis. Also, define omission and commission in

the figure caption.

11. P.12, line 8: Delete ")" at the end of the sentence.

12. P.12, line 21, should it be "rates" instead of "numbers"?

13. P.13, line 22 and p.14, Figure 7: The dots on the figure represent each sample of data. Can you indicate how many samples for each dataset used?

14. p.15, Table 1: It could also be a barplot.

---

## Referee Comment (RC2) · F. Lemonnier (Referee) · 17 Jul 2018

Dear Dr. Philip Marsh,

I have reviewed the manuscript "Evaluation of the CloudSat surface snowfall product over Antarctica using ground-based precipitation radars" by Niels Souverijns and colleagues, submitted for publication in The Cryosphere.

The paper explores various parameters of the CloudSat snowfall climatology proposed by Palerme et al., 2014, such as its temporal sampling rate and its spatial resolution. This climatology is evaluated by way of a comparison with observations from three

different ground micro-rain radars. It is also compared with ERA-Interim reanalysis, which is designated as a reference in regards with the simulated Antarctic snowfall. The authors conclude that the CloudSat snowfall climatology, at a resolution of $1^o$ latitude by $2^o$ longitude, represents well the snowfall climatology of each MRR site and is more effective than ERA-Interim reanalysis, but cannot be considered for individual snowfall events.

The topic of the paper is certainly appropriate for The Cryosphere, and assesses the CloudSat climatology as an effective tool for validating climate models. The manuscript is presented clearly, however, after reviewing this article, I have a few scientific questions that I will explain below.

Sincerely,

Florentin Lemonnier

**Science questions**

- Page 5, $15^{th}$ line. It is mentioned that the difference between CloudSat (1200 m a.g.l.) and the MRRs (300 m a.g.l) is valued by 9-11%, according to Maahn et al., 2014, at the PE station while at DDU it equals 13%. According to recent studies (such as Grazioli et al., 2017b), coastal areas, such as the DDU and MZ stations are blown by sudden strong katabatic winds. The authors could have compared snowfall rates at the vertical MRR level corresponding to CloudSat first bin. Afterwards they could have evaluated the discrepancies of each MRR between 1200 m and 300 m a.g.l. by studying their vertical profiles, instead of considering an estimated value of the gap between CloudSat and ground radars.

- Page 5, $20^{th}$ line. The difference in snowfall rate between the first bin of the MRRs and the surface is not considered in this study. It has been simulated by ECMWF IFS (Grazioli et al., 2017b) that 35% of the snowfall is sublimating in

the lower kilometer of the atmosphere over the Nov-2015 to Oct-2016 period, where the surface is lower than 1 km above sea level. By studying the average vertical profiles of each MRR over their corresponding periods of observation, can the authors establish a trend from this sublimation to the surface, quantify it and estimate its effect on their ground snowfall estimations ?

- Page 13, $27^{th}$ line. When the authors mention that "CloudSat is not able to capture individual snowfall events adequately at a single location", I think the authors should be more specific about that assertion. Indeed for specific precipitation cases, when the satellite overpasses a station closely, if the ground-radar and the CloudSat radar are properly calibrated and their Ze-Sr relations well-established, they should capture a similar precipitation rate.

- Page 14, $1^{st}$ line. ERA-Interim reanalysis provides surface snowfall. Is it relevant to compare this surface product with 1200 m a.g.l and 300 m a.g.l observations ? Do you take into account the effects of the low level sublimation processes on the first bin CloudSat and the first bin MRR measurements ?

---

## Author Comment (AC1) · 4 Sep 2018

**Response to Reviewer 1 Comments:**
**Evaluation of the CloudSat surface snowfall product over Antarctica using ground-based precipitation radars**

Niels Souverijns, Alexandra Gossart, Stef Lhermitte, Irina V. Gorodetskaya,
Jacopo Grazioli, Claudio Duran-Alarcon, Brice Boudevillain,
Christophe Genthon, Claudio Scarchilli and Nicole P.M. van Lipzig

September 4, 2018

For clarifying our answers to the reviewers' comments, the following color scheme is used: comments of the reviewer are denoted in blue, our answers are denoted in black and quotes from the revised text are in green.

Before addressing the comments of the reviewer, it must be noted that during the revision process there was detected that a small part of erroneous MRR data at the PE station was included in the analysis. This erroneous data was recorded during the 2015-2016 austral winter season and was caused by interference from other instruments. It was removed from the sample lowering the period of concurrent data availability of the MRR and CloudSat from 928 to 851 days for the PE station (Fig. 2 in the main paper). This mainly affects Fig. 6 in the main paper where a clear lowering of both the MRR and CloudSat total precipitation amount is observed. However, as the total snowfall amount for both the MRR and CloudSat lowered with an equal amount, results and conclusions are not affected significantly.

*This study evaluates the surface snowfall measured derived by CloudSat by comparing it with the product from ERA and Micro Rain Radar signals over Antarctica. The study brings valuable information to the scientific community to better understand changes in snow cover over Antarctica. It uses a combination of ground instruments, satellite and reanalysis data. It was found that CloudSat does not measure well snowfall because of the lower time resolution of passes of the regional whereas ERA covers the entire continent with 3 hourly outputs. The snowfall measured by the MRR is similar to ERA because it can capture single events as opposed to CloudSat. This study contributes to better understand the evolution of snow cover in Antarctica. Finally, the English and the clarity of some figures should also be improved.*

The reviewer is thanked for the review of our manuscript. Improvements regarding the clarity of the figures and the language are discussed in the individual comments below.

*1. P.3, line 27-31: It would be useful to explain in an Appendix the choice of the Z-S relationship chosen and how it was compared to other available data used to measure snowfall at each site (other than CloudSat). Also, could you*

*clarify in the manuscript how the 3 sets of data were compared? If I understood correctly they don't provide the snowfall information at the same altitude. For example, it is indicated in the manuscript that the lowest level where the snowfall is estimated by CloudSat is at 1200m and the MRRs is at 300m above ground level. I assumed that ERA snowfall is probably produced at the surface. How these differences affect the results of your study?*

We expanded the information regarding the choice of the Z-S relationship for each of the stations. There is now specified which other instruments were available at the stations to measure snowfall amounts and which were used for the derivation of the MRR snowfall rates.

*Radar reflectivity measurements were subsequently converted to snowfall rates using relations specifically developed for the MRR at the PE and DDU station. At the PE station, this relation was constructed using information about the snowflake microphysics obtained from a video disdrometer (details in Souverijns et al., 2017), while at the DDU station, the relation was derived based on a weighing gauge, polarimetric radar and snowflake camera (details in Grazioli et al., 2017a).*

We decided not to include more information in an appendix section, as the two publications that were cited (i.e. Souverijns et al., 2017; Grazioli et al., 2017a) are fully devoted to the construction of the Z-S relation for the PE and DDU station including a discussion of snowfall amounts measured by the different instruments. A detailed description is available in these publications.

Regarding the comparison of CloudSat, the MRRs and ERA-Interim, it is indeed true that all instruments are measuring at different height levels. The goal of this paper is mainly to evaluate the performance of the CloudSat snowfall product as an estimator of the surface snowfall amount, which is the currently the main use of the product in the cryospheric community. This is the reason why we have evaluated the CloudSat product (at 1200m a.g.l.) against the MRRs (300m a.g.l.) as these are the closest observations of snowfall currently available over the AIS. ERA-Interim (which is a surface snowfall product) was chosen as this product is currently mainly used for Antarctic-wide surface snowfall estimates. We have clarified our goal in the main text.

*The main interest of the paper is to evaluate the CloudSat snowfall product as an estimate of the surface snowfall amount, which is the primary application for both the observing and modelling community. As such, the lowest usable measurement bin of both instruments is considered in the analysis.*

*The CloudSat snowfall climatology provides very good results compared to MRR total snowfall amount records for all three stations, showing the skill of CloudSat for the estimation of the surface snowfall climatology over the AIS, outperforming ERA-Interim reanalysis.*

As such, the main problem is that the lowest bin of the MRR cannot be considered ground-truth and that sublimation can occur between 300m a.g.l. and the surface.

For the PE station, the amount of sublimation between the lowest measurement bin of the MRR and the surface was calculated using the height correction of Wood (2011), by extrapolating the trend in the lowest MRR vertical levels towards the surface to account for horizontal displacement and sublimation. This resulted in an average decrease of radar reflectivity of 1.66

dBz in case sublimation was detected in the lowest bins of the MRR (Souverijns et al., 2017) and would lead to an overestimation of the snowfall rate by 29 %. As this correction was only applied during events with a clear sublimation signal (approximately 15 % of the precipitation events), the impact on the total snowfall amount is limited.

For the DDU station, three model simulations have been performed simulating the vertical profile of precipitation. Based on the results of Fig. 2 of Grazioli et al. (2017b), two models predict an overestimation of 7 % of the cumulative snowfall record at the 300m a.g.l. level compared to the surface.

The vertical profile of precipitation measured by the MRRs is given in Fig. S3 (Fig. R1 in this document) for the three stations for the periods of concurrent measurements with CloudSat. It is possible to extrapolate the trend from the lowest measurement bins towards the surface using a similar approach as applied in (Wood, 2011; Souverijns et al., 2017). This leads to an overestimation of 14 % of the total snowfall amount at 300m a.g.l. compared to the surface for the PE station, 9 % for the DDU station and 7 % for the MZ station. These numbers are in line with the results of Souverijns et al. (2017); Grazioli et al. (2017b) for the PE and DDU station respectively discussed also above. The difference in numbers for the PE station between this study and Souverijns et al. (2017) can be attributed to the fact that different time periods are studied.

[Figure]

Figure R1: Total snowfall amount as a function of height above ground level as obtained by the MRRs for the periods of concurrent measurements depicted in Fig. 2 of the main paper.

The description of sublimation in the lowest layers of the atmosphere is expanded in the main text.

*Furthermore, sublimation persists towards the surface, also influencing the layer between the lowest measurement bin of the MRR (i.e. 300m a.g.l.) and the surface, where typically an inversion and katabatic flow is present (Grazioli et al., 2017b; Souverijns et al., 2017). The amount of sublimation in the lowest 300m of the atmosphere can be calculated by extrapolating the vertical trend in snowfall rates towards the surface following the approach of Wood (2011) leading to an overestimation of the snowfall rate at 300m a.g.l. of 14 %, 9 % and 7 % for respectively the PE, DDU and MZ station compared to the surface. One must note that sublimation increases the saturation level of the atmosphere, negatively influencing future sublimation. Therefore, the method of Wood (2011) might overestimate the amount of sublimation. The discrepancy in the lowest 300m of the atmosphere is not considered in this study but needs to be accounted for.*

Based on Fig. R1, a large discrepancy is detected between MRR snowfall rates at 300m

a.g.l. and 1200m a.g.l.. Despite not being the main goal of this study, the difference is investigated in more detail and a comparison between the MRR snowfall rates at 1200m a.g.l. and CloudSat is executed.

Fig. 6 & 7 from the main paper are reproduced for MRR measurements at 1200m a.g.l. (Fig. S1 & S2 in the Supplement; Fig. R2 & R3 in this document). A lowering of the total MRR snowfall amount is observed for all stations. For the PE station, a 26% decrease in total snowfall amounts is observed. This value is much larger than the number obtained by Maahn et al. (2014) which only found a decrease rate of 11%. The discrepancy between both values can be attributed to the lack of data availability in the study of Maahn et al. (2014). There, only one full year of MRR measurements was available, namely 2012. In 2012, no heavy snowfall events were recorded with precipitation rates exceeding 1 mm/h. In our study, data from 2010-2016 was included. During this longer time period, several large events (> 5 mm/h) were recorded. An overview of the total snowfall amount as a function of height is added to the Supplement (Fig. S3; Fig. R1 in this document). Over the PE station, large snowfall events have the tendency to attribute for large amounts of augmentation in the lowest kilometer of the atmosphere. Furthermore, a distinct number of these large snowfall events have a vertical extent less than 1 km. An example of these types of events are given in Fig. R4. As these events occurred less often in 2012, Maahn et al. (2014) obtained lower values.

[Figure]

Figure R2: (first row) Overview of the total snowfall amounts for the three stations as observed by CloudSat and the Micro Rain Radars during the periods of collocated measurements (Fig. 2 in the main paper). (second row) Individual snowfall event error analysis. As Micro Rain Radar snowfall rates are considered truth, omission errors are defined as an underestimation, while commission errors are an overestimation of snowfall rates by CloudSat. The x-axis denotes different spatial resolutions of the CloudSat climatology (grid box longitudinal resolution = 2 * grid box latitudinal resolution).

For the MZ station, the same amount of precipitation reduction is obtained as for the PE station (25%; Fig. R2). The vertical profile of total precipitation shows that the layer of maxi-

[Figure]

Figure R3: (first row) Empirical cumulative distribution of MRR and CloudSat snowfall events at a spatial resolution of 1° latitude by 2° longitude. (second row) Direct comparison between MRR and CloudSat individual snowfall events. $R^2$ denotes the adjusted coefficient of determination, RMSE is the root mean square error, N indicates the number of observations, while the thin line is the bisector.

[Figure]

Figure R4: Radar reflectivity spectrum for two snowfall events at the PE station (upper: 24 Feb 2015; lower: 22 Dec 2013).

mum precipitation extends up to 700m after which a sharp decrease is found (Fig. R1). Similar precipitation events as found for the PE station and visualised in Fig. R4 have been observed. This leads to the large difference in precipitation amounts between the 300m and 1200m a.g.l. level.

For the DDU station, a reduction in total snowfall amount of 8 % was observed between the

300 and 1200m a.g.l. level (Fig. R2). This low value can be attributed to the fact that precipitation systems at DDU have a much larger vertical extent and highest precipitation numbers are not limited to the lowest layers. As the augmentation layer extents to higher altitudes, a better agreement of snowfall rates between altitudes of 300m and 1200m a.g.l. is obtained.

These results are now referred to in the main text.

*The data acquisition height difference between CloudSat (1200m a.g.l.) and the MRRs (300m a.g.l.) accounts for an average underestimation of 25 % in total snowfall amount by CloudSat compared to the MRR at the PE station. At the DDU station this equals 8 % (Grazioli et al., 2017b), while at the MZ station, an underestimation of 25 % is obtained. A discussion on the source of this discrepancy in snowfall amount between the 300m and 1200m level can be found in the Supplement (Text S1 and Figs. S1-S3).*

It is remarkable that for the PE and MZ station, the comparison between CloudSat and the MRR both measuring at 1200m a.g.l. attributes for less good results compared to MRR measurements at 300m a.g.l. (compare Fig. 6 in the main paper and Fig. R2). This shows that CloudSat overestimates the precipitation amount at 1200m a.g.l. leading to commission errors. CloudSat has a tendency to overestimate the frequency of snowfall events, attributing for the worse performance, even though a better match in the cumulative distribution is obtained (compare Fig. 7 in the main paper and Fig. R3).

*Furthermore, the difference in acquisition height between both instruments is not taken into account in the above analysis. In case the MRR measures snowfall rates at the same level as CloudSat (i.e. 1200m a.g.l.), a significant lower amount of snowfall is recorded. As CloudSat is known to overestimate the frequency of small snowfall events (Chen et al., 2016), this can be interpreted as an extra source of commission errors, although a better match in the cumulative distribution is achieved. A thorough discussion on this discrepancy can be found in the Supplement (Text S1 and Figs. S1-S3).*

As some interesting new insights are obtained, the text discussing these issues was added to the Supplement.

*Apart from evaluating the CloudSat snowfall climatology and individual events (obtained at 1200m a.g.l.) with MRR measurements at the level closest to the surface (300m a.g.l.), an extra comparison is executed by including MRR measurements at 1200m a.g.l.. The higher level of snowfall rate acquisition of the MRR leads to a decrease in the total snowfall amount of 26 %, 8 % and 25 % for respectively the PE, DDU and MZ station compared to measurements at 300m a.g.l. (compare Fig. S1 and Fig. 6 in the main paper). The total snowfall amount as a function of height is visualised in Fig. S3 and is characterised with a typical shape for all stations. Highest snowfall rates are usually obtained a few hundreds meter above the surface. Towards the surface lower values are observed, induced by katabatic winds that cause sublimation (Grazioli et al., 2017b). The decrease towards higher altitudes is governed by the vertical extent of the precipitation systems, which are often present only in the lowest layers of the atmosphere (Maahn et al., 2014). For the PE and MZ station, larger discrepancies between the 300m and 1200m a.g.l. level are obtained. This can be attributed to the fact that for these stations, highest precipitation intensities are mainly located below 700m a.g.l., indicating that the vertical extent of the precipitation systems is generally low for these stations (Fig. S3). For the DDU station, precipitation systems usually have a larger vertical extent. Therefore, the steady decrease in snowfall rates for higher altitudes only starts from heights over 1000m a.g.l.,*

*attributing for the minor differences in snowfall rates between the 300m and 1200m a.g.l. level for this station.*

*The lower total amount of snowfall rates obtained at 1200m a.g.l. by the MRRs leads, counter-intuitively, to worse performances compared to the snowfall rates obtained by CloudSat at 1200m a.g.l. for both the PE and MZ station (compare Fig. S1 and Fig. 6 in the main paper). When investigating the cumulative distribution of snowfall rates obtained by both instruments, a better agreement is obtained for both stations compared to the initial assessment using MRR measurements at 300m a.g.l. (compare Fig. S2 and Fig. 7 in the main paper). The main reason for the overestimation of CloudSat snowfall rates compared to MRR snowfall rates at 1200m a.g.l. is therefore attributed to the much higher frequency of snowfall events detected in CloudSat (Chen et al., 2016), leading to high commission errors. In the comparison at 300m a.g.l., this overestimation of the frequency of snowfall events was compensated by the higher snowfall rates registered by the MRR (omission errors; Fig. 7 in the main paper), which is not the case at 1200m a.g.l. (Fig. S2). For the DDU station, the frequency of snowfall event detection is approximately equal, explaining the better performance for this station.*

**2. P.4, Figure 1: Could you add the name of the stations on the map with the acronym used in the text and other figures?**

We added the acronym of the stations to the title of each of the DEMs (Figure R5).

[Figure]

Figure R5: Digital Elevation Map of the Antarctic Ice Sheet (Liu et al., 2015) with three insets corresponding to the location of the Micro Rain Radars. Upper: Princess Elisabeth station (PE), right: Mario Zucchelli station (MZ), lower: Dumont D'Urville station (DDU). The inset at the bottom left shows the Micro Rain Radar at the Princess Elisabeth station.

**3. P.5, line 25, Please double check the guidelines for references. The reference should be (conforme Palerme et al., 2014) instead of (conforme Palerme et al. (2014)).**

This is indeed correct. We adapted the reference accordingly. Furthermore, this issue has been adapted in one other place in the manuscript.

*... , mainly driven by large-scale circulation (i.e. cyclonic activity in the circumpolar trough; Gorodetskaya et al., 2013, 2014; Souverijns et al., 2018).*

**4. P.6, line 5, How is a snowfall event defined? Were they defined per day or per time period when snow accumulated at the ground (or at the lowest MRR level)?**

Regarding the comparison of individual snowfall events detected by both the MRR and CloudSat, events are rigorously defined. As the events need to be detected by both instruments, we are restricted to the CloudSat overpasses. During a CloudSat overpass close by the station, a spatial area within the grid box of 1° latitude by 2° longitude is covered by its track. The distance of this track within the grid box is converted to a time period, i.e. if the track is 130 km long within the grid box and the wind speed at 300m a.g.l. (which is obtained from ERA-Interim reanalysis data over the stations (Dee et al., 2011)) equals 20 km h$^{-1}$, the MRR subsample covers a time period of 6.5 hours. The definition of this comparison period is explained in the result section.

*Each of these MRR subsamples however needs to cover a time period to obtain a fair estimate of the temporal uncertainty induced by the CloudSat temporal revisit time. CloudSat has a narrow swath width. During a CloudSat overpass close by the station, a spatial area within the grid box of 1° latitude by 2° longitude is covered by its track (see Sect. 2.2). The distance of this track within the grid box is converted to a time period, i.e. if the track is 130 km long within the grid box and the wind speed at 300m a.g.l. (which is acquired from ERA-Interim reanalysis data over the stations (Dee et al., 2011)) equals 20 km h$^{-1}$, the MRR subsample covers a time period of 6.5 hours.*

*In order to facilitate the comparison, MRR snowfall rates are calculated by averaging snowfall rates over a time period following the same procedure as in Sect. 3.1. This time period depends on the spatial resolution of the grid and the wind speed at 300m a.g.l.. For example, if the grid has a spatial resolution of 1° latitude by 2° longitude (i.e. with a maximal distance of 130 km between the edges of the grid box) and the wind speed equals 20 km h$^{-1}$, the MRR record is averaged over 6.5 hours. The minimal MRR averaging period is one hour). Using this methodology, one has to assume that the precipitation systems are stationary in time and uniform in space, which is not valid over highly variable topography (see Sect. 2.3). This source of error needs to be considered when comparing both instruments.*

**5. P.7, line 16-26, there are 3 times "in order to get" in the same paragraph. It should be reworded.**

We have adapted the paragraph so "in order to get" is limited to the first sentence only.

*In order to get an estimate of the uncertainty induced by the low temporal sampling frequency of CloudSat, systematic sampling is applied on the MRR snowfall record (available on the minute time-scale). For the MZ station for example, the revisit time equals approximately 2.1 days. As such, subsamples are extracted from the MRR record with an interval of 2.1 days. Each of these MRR subsamples however needs to cover a time period to obtain a fair estimate of the temporal uncertainty induced by the CloudSat temporal revisit time. CloudSat has a narrow swath width. During a CloudSat overpass close by the station, a spatial area within the grid box of 1° latitude by 2° longitude is covered by its track (see Sect. 2.2). The distance of this track within the grid box is converted to a time period, i.e. if the track is 130 km long within the grid box and the wind speed at 300m a.g.l. (which is acquired from ERA-Interim reanalysis data over the stations (Dee et al., 2011)) equals 20 km h$^{-1}$, the MRR subsample covers a time*

*period of 6.5 hours. On average, this time period equals 7.2, 7.4 and 6.9 hours respectively for the PE, DDU and MZ station. As such, in case of the example for the MZ station, for each bootstrap a subsample of 6.9 hours is extracted every 2.1 days as a means to obtain a correct estimation of the CloudSat temporal uncertainty (Fig. 5).*

**6. P.8, Figure 3, It is surprising to see that the MRR misses many events detected by ERA. It should be further discussed in the manuscript. Also, a legend should be added to the figure.**

We added a discussion of the comparison between ERA-Interim and the MRRs.

*For all stations, ERA-Interim reanalysis underestimates the snowfall amount of large events, which has also been observed in Fig. 3, attaining for omission errors similar to CloudSat (see Sect. 3.2). This underestimation is related to the fact that high peaks in snowfall are smoothed out over the grid. Smaller snowfall events are much better captured by ERA-Interim compared to CloudSat (see also Fig. 3 & 4). However, a substantial number of small events are detected in ERA-Interim that were not registered by the MRRs, mainly for PE and MZ station (Fig. 8). This can be related to the topography of the surroundings, leading to localised snowfall, which is gridded to low resolution data products as ERA-Interim and/or other sources as e.g. erroneous erroneous moisture fluxes.*

Furthermore, we added a legend to Fig. 3 (Fig. R6 in this document).

[Figure]

Figure R6: Snowfall rates (mm w.e. h$^{-1}$) during March 2016 at the three stations derived from the MRRs (blue bars), the grid box comprising each of the three stations in ERA-Interim reanalysis (green) and the average of the CloudSat overpasses in the grid box (1° latitude by 2° longitude) comprising each of the three stations following the approach of Palerme et al. (2014) (red).

*7. P.9, Figure 4, Is it possible to do the same figure from ERA and maybe CloudSat? It would be interesting to see how well they compare among each other if possible.*

We have created the same figure for CloudSat (Fig. R7) and ERA-Interim (Fig. R8).

[Figure]

(a) Princess Elisabeth station    (b) Dumont D'Urville station    (c) Mario Zucchelli station

Figure R7: Seasonal variability of snowfall amounts derived from CloudSat at the three stations.

[Figure]

(a) Princess Elisabeth station    (b) Dumont D'Urville station    (c) Mario Zucchelli station

Figure R8: Seasonal variability of snowfall amounts derived from ERA-Interim at the three stations.

For CloudSat this visualisation does not have much added value due to the lack of observations. Only a limited number of snowfall events are detected over the stations (69, 26 and 47 for the PE, DDU and MZ station respectively). When these are subdivided in seasons, no clear distribution can be deduced due to a lack of large snowfall events.

For ERA-Interim, a similar seasonal cycle is detected compared to the MRR results (compare Fig. R8 & Fig. 4 in the main paper) and no big discrepancies are found. The main difference is the high frequency of small snow storms observed in austral summer (DJF) over DDU compared to results of the MRR. For all stations, ERA-Interim overestimates the frequency of small snowfall events, while underestimating the frequency of large snowfall events. This has also been described and concluded from the results in section 3.3 and Fig. 7 in the main paper. We have added the results of ERA-Interim (Fig. R8) to Fig. 4 in the main paper and included several references to the text referring to this figure.

*It is noted that precipitation observations in winter are scarce for the MRR (Sect. 2.3), while interannual precipitation variability can be large. At the PE and MZ stations, snowfall events of highest intensities are limited to the austral spring (SON) and summer season, while during austral winter, lighter snowfall events are recorded in both the MRR and ERA-Interim record. This complies with van Lipzig et al. (2002) in their study of the seasonality of the SMB*

*over Dronning Maud Land. For the DDU station, a larger number of high-intensity snowfall events are observed. Seasonally, at DDU the lowest snowfall amounts are obtained during austral summer, while highest contributions to the total snowfall record are obtained during the other months, confirming the results of Grazioli et al. (2017a). In the ERA-Interim record, the opposite result is obtained, showing a peak in low intensity snowfall events during austral summer. A clear discrepancy in frequencies is observed between the MRR and ERA-Interim snowfall record for all stations. ERA-Interim detects more low-intensity snowfall events, while underestimating the amount of high-intensity storms. This inconsistency is further elaborated in Sect. 3.3.*

*For all stations, ERA-Interim reanalysis underestimates the snowfall amount of large events, which has also been observed in Fig. 4, attaining for omission errors similar to CloudSat (see Sect. 3.2). This underestimation is related to the fact that high peaks in snowfall are smoothed out over the grid. Smaller snowfall events are much better captured by ERA-Interim compared to CloudSat (see also Fig. 3 & 4).*

**8. P.9, line 9, Should it be Figure 4 instead of Figure 7?**

This is indeed a typo. The text has been adapted accordingly.

**9. P.10, Figure 5: Add the title of the y-axis on the left column.**

We have adapted the figure to increase its readability (Fig. R9 in this document). As we flipped the figure, the names of the stations are now displayed at the top of the figure. Furthermore, the y-axis label has been added.

**10. P.11, Figure 6: Add the title of the y-axis. Also, define omission and commission in the figure caption.**

The label of the y-axis has been adapted to also include a description. Furthermore, we have added a definition of omission and commission errors to the figure caption (Fig. R10 in this document).

**11. P.12, line 8: Delete ")" at the end of the sentence.**

This is indeed a typo. The sentence has been adapted accordingly.

**12. P.12, line 21, should it be "rates" instead of "numbers"?**

This is a correct remark of the referee. We have adapted the sentence in order to correctly represent the features described herein.

*As the distribution of snowfall rates is skewed towards high intensities (Fig. 4 in the main paper), these snowfall events are missed leading to an underestimation of the total snowfall amount, which is indeed observed for all stations (Fig. 6 in the main paper).*

**13. P.13, line 22 and p.14, Figure 7: The dots on the figure represent each sample of data. Can you indicate how many samples for each dataset used?**

The number of samples were added to the figure for each of the stations (Fig. R11 in this

[Figure]

Figure R9: Boxplots showing the uncertainty when applying systematic sampling on the MRR snowfall record (10.000 bootstraps) using different temporal sampling frequencies (x-axis, D denotes days). Total snowfall amounts during collocated periods of MRR and CloudSat measurements (top) and the 95[th] percentile snowfall rate (bottom) are shown. The bottom and top edges of the boxplot indicate the 25-75[th] percentile (dark pink shading), while the whiskers denote the 10-90[th] percentile (light pink shading). The red line denotes the median.

document). Furthermore, we also added the number of samples to Fig. 9 in the main paper (Fig. R12 in this document).

**14. p.15, Table 1: It could also be a barplot.**

We have replaced the table by a bar plot visualisation and added it to the manuscript (Fig. R13 in this document). The table has been moved to the Supplement.

[revised manuscript text omitted]

---

## Author Comment (AC2) · 4 Sep 2018

**Response to Reviewer 2 Comments:**
**Evaluation of the CloudSat surface snowfall product over Antarctica using ground-based precipitation radars**

Niels Souverijns, Alexandra Gossart, Stef Lhermitte, Irina V. Gorodetskaya,
Jacopo Grazioli, Claudio Duran-Alarcon, Brice Boudevillain,
Christophe Genthon, Claudio Scarchilli and Nicole P.M. van Lipzig

September 4, 2018

For clarifying our answers to the reviewers' comments, the following color scheme is used: comments of the reviewer are denoted in blue, our answers are denoted in black and quotes from the revised text are in green.

Before addressing the comments of the reviewer, it must be noted that during the revision process there was detected that a small part of erroneous MRR data at the PE station was included in the analysis. This erroneous data was recorded during the 2015-2016 austral winter season and was caused by interference from other instruments. It was removed from the sample lowering the period of concurrent data availability of the MRR and CloudSat from 928 to 851 days for the PE station (Fig. 2 in the main paper). This mainly affects Fig. 6 in the main paper where a clear lowering of both the MRR and CloudSat total precipitation amount is observed. However, as the total snowfall amount for both the MRR and CloudSat lowered with an equal amount, results and conclusions are not affected significantly.

*The paper explores various parameters of the CloudSat snowfall climatology proposed by Palerme et al. (2014), such as its temporal sampling rate and its spatial resolution. This climatology is evaluated by way of a comparison with observations from three different ground micro-rain radars. It is also compared with ERA-Interim reanalysis, which is designated as a reference in regards with the simulated Antarctic snowfall. The authors conclude that the CloudSat snowfall climatology, at a resolution of 1° latitude by 2° longitude, represents well the snowfall climatology of each MRR site and is more effective than ERA-Interim reanalysis, but cannot be considered for individual snowfall events. The topic of the paper is certainly appropriate for The Cryosphere, and assesses the CloudSat climatology as an effective tool for validating climate models. The manuscript is presented clearly, however, after reviewing this article, I have a few scientific questions that I will explain below.*

We thank the reviewer for the review of the manuscript. The specific comments are addressed below.

*Page 5, 15th line. It is mentioned that the difference between CloudSat*

*(1200m a.g.l.) and the MRRs (300m a.g.l) is valued by 9-11%, according to Maahn et al. (2014) at the PE station while at DDU it equals 13%. According to recent studies (such as Grazioli et al. (2017)), coastal areas, such as the DDU and MZ stations are blown by sudden strong katabatic winds. The authors could have compared snowfall rates at the vertical MRR level corresponding to Cloud-Sat first bin. Afterwards they could have evaluated the discrepancies of each MRR between 1200m and 300m a.g.l. by studying their vertical profiles, instead of considering an estimated value of the gap between CloudSat and ground radars.*

The goal of this paper is mainly to evaluate the performance of the CloudSat snowfall product as an estimator of the surface snowfall amount, which is the currently the main use of the product in the cryospheric community. This is the reason why we have evaluated the CloudSat product (at 1200m a.g.l.) against the MRRs (300m a.g.l.) as these are the closest observations of snowfall currently available over the AIS. We have clarified our goal in the main text.

*The main interest of the paper is to evaluate the CloudSat snowfall product as an estimate of the surface snowfall amount, which is the primary application for both the observing and modelling community. As such, the lowest usable measurement bin of both instruments is considered in the analysis.*

*The CloudSat snowfall climatology provides very good results compared to MRR total snowfall amount records for all three stations, showing the skill of CloudSat for the estimation of the surface snowfall climatology over the AIS, outperforming ERA-Interim reanalysis.*

It is acknowledged that this approach includes several deficiencies. As stated by the reviewer and observed by Maahn et al. (2014) for the PE station and Grazioli et al. (2017) for the DDU station, there can be a large discrepancy between the snowfall rates obtained at the CloudSat and MRR acquisition level. It is therefore appropriate to also investigate these differences in this paper and to not only rely on the results of previous work to gain more insight in the performance of CloudSat and the MRR at the same height acquisition level. As such, part of the analysis was repeated using MRR snowfall rates acquired at the 1200m a.g.l. measurement bin.

Fig. 6 & 7 from the main paper are reproduced for MRR measurements at 1200m a.g.l. (Fig. S1 & S2 in the Supplement; Fig. R1 & R2 in this document). A lowering of the total MRR snowfall amount is observed for all stations. For the PE station, a 26% decrease in total snowfall amounts is observed. This value is much larger than the number obtained by Maahn et al. (2014) which only found a decrease rate of 11%. The discrepancy between both values can be attributed to the lack of data availability in the study of Maahn et al. (2014). There, only one full year of MRR measurements was available, namely 2012. In 2012, no heavy snowfall events were recorded with precipitation rates exceeding 1 mm/h. In our study, data from 2010-2016 was included. During this longer time period, several large events ($> 5$ mm/h) were recorded. An overview of the total snowfall amount as a function of height is added to the Supplement (Fig. S3; Fig. R3 in this document). Over the PE station, large snowfall events have the tendency to attribute for large amounts of augmentation in the lowest kilometer of the atmosphere. Furthermore, a distinct number of these large snowfall events have a vertical extent less than 1 km. An example of these types of events are given in Fig. R4. As these events occurred less often in 2012, Maahn et al. (2014) obtained lower values.

For the MZ station, the same amount of precipitation reduction is obtained as for the PE

[Figure]

Figure R1: (first row) Overview of the total snowfall amounts for the three stations as observed by CloudSat and the Micro Rain Radars during the periods of collocated measurements (Fig. 2 in the main paper). (second row) Individual snowfall event error analysis. As Micro Rain Radar snowfall rates are considered truth, omission errors are defined as an underestimation, while commission errors are an overestimation of snowfall rates by CloudSat. The x-axis denotes different spatial resolutions of the CloudSat climatology (grid box longitudinal resolution = 2 * grid box latitudinal resolution).

station (25%; Fig. R1). The vertical profile of total precipitation shows that the layer of maximum precipitation extends up to 700m after which a sharp decrease is found (Fig. R3). Similar precipitation events as found for the PE station and visualised in Fig. R4 have been observed. This leads to the large difference in precipitation amounts between the 300m and 1200m a.g.l. level.

For the DDU station, a reduction in total snowfall amount of 8 % was observed between the 300 and 1200m a.g.l. level (Fig. R1). This low value can be attributed to the fact that precipitation systems at DDU have a much larger vertical extent and highest precipitation numbers are not limited to the lowest layers. As the augmentation layer extents to higher altitudes, a better agreement of snowfall rates between altitudes of 300m and 1200m a.g.l. is obtained.

These results are now referred to in the main text.

*The data acquisition height difference between CloudSat (1200m a.g.l.) and the MRRs (300m a.g.l.) accounts for an average underestimation of 25 % in total snowfall amount by CloudSat compared to the MRR at the PE station. At the DDU station this equals 8 % (Grazioli et al., 2017), while at the MZ station, an underestimation of 25 % is obtained. A discussion on the source of this discrepancy in snowfall amount between the 300m and 1200m level can be found in the Supplement (Text S1 and Figs. S1-S3).*

It is remarkable that for the PE and MZ station, the comparison between CloudSat and

[Figure]

Figure R2: (first row) Empirical cumulative distribution of MRR and CloudSat snowfall events at a spatial resolution of 1° latitude by 2° longitude. (second row) Direct comparison between MRR and CloudSat individual snowfall events. $R^2$ denotes the adjusted coefficient of determination, RMSE is the root mean square error, N indicates the number of observations, while the thin line is the bisector.

[Figure]

Figure R3: Total snowfall amount as a function of height above ground level as obtained by the MRRs for the periods of concurrent measurements depicted in Fig. 2 of the main paper.

the MRR both measuring at 1200m a.g.l. attributes for less good results compared to MRR measurements at 300m a.g.l. (compare Fig. 6 in the main paper and Fig. R1). This shows that CloudSat overestimates the precipitation amount at 1200m a.g.l. leading to commission errors. CloudSat has a tendency to overestimate the frequency of snowfall events, attributing for the worse performance, even though a better match in the cumulative distribution is obtained (compare Fig. 7 in the main paper and Fig. R2).

*Furthermore, the difference in acquisition height between both instruments is not taken into account in the above analysis. In case the MRR measures snowfall rates at the same level as CloudSat (i.e. 1200m a.g.l.), a significant lower amount of snowfall is recorded. As CloudSat is known to overestimate the frequency of small snowfall events (Chen et al., 2016), this can be interpreted as an extra source of commission errors, although a better match in the cumulative distribution is achieved. A thorough discussion on this discrepancy can be found in the Supple-*

[Figure]

Figure R4: Radar reflectivity spectrum for two snowfall events at the PE station (upper: 24 Feb 2015; lower: 22 Dec 2013).

*ment (Text S1 and Figs. S1-S3).*

As some interesting new insights are obtained, a text discussing these issues was added to the Supplement.

*Apart from evaluating the CloudSat snowfall climatology and individual events (obtained at 1200m a.g.l.) with MRR measurements at the level closest to the surface (300m a.g.l.), an extra comparison is executed by including MRR measurements at 1200m a.g.l.. The higher level of snowfall rate acquisition of the MRR leads to a decrease in the total snowfall amount of 26 %, 8 % and 25 % for respectively the PE, DDU and MZ station compared to measurements at 300m a.g.l. (compare Fig. S1 and Fig. 6 in the main paper). The total snowfall amount as a function of height is visualised in Fig. S3 and is characterised with a typical shape for all stations. Highest snowfall rates are usually obtained a few hundreds meter above the surface. Towards the surface lower values are observed, induced by katabatic winds that cause sublimation (Grazioli et al., 2017). The decrease towards higher altitudes is governed by the vertical extent of the precipitation systems, which are often present only in the lowest layers of the atmosphere (Maahn et al., 2014). For the PE and MZ station, larger discrepancies between the 300m and 1200m a.g.l. level are obtained. This can be attributed to the fact that for these stations, highest precipitation intensities are mainly located below 700m a.g.l., indicating that the vertical extent of the precipitation systems is generally low for these stations (Fig. S3). For the DDU station, precipitation systems usually have a larger vertical extent. Therefore, the steady decrease in snowfall rates for higher altitudes only starts from heights over 1000m a.g.l., attributing for the minor differences in snowfall rates between the 300m and 1200m a.g.l. level for this station.*

*The lower total amount of snowfall rates obtained at 1200m a.g.l. by the MRRs leads, counter-intuitively, to worse performances compared to the snowfall rates obtained by CloudSat at 1200m a.g.l. for both the PE and MZ station (compare Fig. S1 and Fig. 6 in the main paper). When investigating the cumulative distribution of snowfall rates obtained by both instruments, a better agreement is obtained for both stations compared to the initial assessment using MRR measurements at 300m a.g.l. (compare Fig. S2 and Fig. 7 in the main paper).*

*The main reason for the overestimation of CloudSat snowfall rates compared to MRR snowfall rates at 1200m a.g.l. is therefore attributed to the much higher frequency of snowfall events detected in CloudSat (Chen et al., 2016), leading to high commission errors. In the comparison at 300m a.g.l., this overestimation of the frequency of snowfall events was compensated by the higher snowfall rates registered by the MRR (omission errors; Fig. 7 in the main paper), which is not the case at 1200m a.g.l. (Fig. S2). For the DDU station, the frequency of snowfall event detection is approximately equal, explaining the better performance for this station.*

**Page 5, 20th line. The difference in snowfall rate between the first bin of the MRRs and the surface is not considered in this study. It has been simulated by ECMWF IFS (Grazioli et al., 2017) that 35% of the snowfall is sublimating in the lower kilometer of the atmosphere over the Nov-2015 to Oct-2016 period, where the surface is lower than 1 km above sea level. By studying the average vertical profiles of each MRR over their corresponding periods of observation, can the authors establish a trend from this sublimation to the surface, quantify it and estimate its effect on their ground snowfall estimations?**

It is indeed noted that the lowest bin of the MRR cannot be considered ground-truth and that significant amounts of sublimation can occur between 300m a.g.l. and the surface. This is a drawback of the study which needs to be considered by the reader.

For the PE station, the amount of sublimation between the lowest measurement bin of the MRR and the surface was calculated using the height correction of Wood (2011), by extrapolating the trend in the lowest MRR vertical levels towards the surface to account for horizontal displacement and sublimation. This resulted in an average decrease of radar reflectivity of 1.66 dBz in case sublimation was detected in the lowest bins of the MRR (Souverijns et al., 2017) and would lead to an overestimation of the snowfall rate by 29 %. As this correction was only applied during events with a clear sublimation signal (approximately 15 % of the precipitation events), the impact on the total snowfall amount is limited.

For the DDU station, three model simulations have been performed simulating the vertical profile of precipitation. Based on the results of Fig. 2 of Grazioli et al. (2017), two models predict an overestimation of 7 % of the cumulative snowfall record at the 300m a.g.l. level compared to the surface.

As the reviewer suggests, it is possible to extrapolate the trend from the lowest measurement bins towards the surface using a similar approach as applied in (Wood, 2011; Souverijns et al., 2017). This leads to an overestimation of 14 % of the total snowfall amount at 300m a.g.l. compared to the surface for the PE station, 9 % for the DDU station and 7 % for the MZ station. These numbers are in line with the results of Souverijns et al. (2017); Grazioli et al. (2017) for the PE and DDU station respectively discussed also above. The difference in numbers for the PE station between this study and Souverijns et al. (2017) can be attributed to the fact that different time periods are studied.

The description of sublimation in the lowest layers of the atmosphere is expanded in the main text.

*Furthermore, sublimation persists towards the surface, also influencing the layer between the lowest measurement bin of the MRR (i.e. 300m a.g.l.) and the surface, where typically an inversion and katabatic flow is present (Grazioli et al., 2017; Souverijns et al., 2017). The amount of*

*sublimation in the lowest 300m of the atmosphere can be calculated by extrapolating the vertical trend in snowfall rates towards the surface following the approach of Wood (2011) leading to an overestimation of the snowfall rate at 300m a.g.l. of 14 %, 9 % and 7 % for respectively the PE, DDU and MZ station compared to the surface. One must note that sublimation increases the saturation level of the atmosphere, negatively influencing future sublimation. Therefore, the method of Wood (2011) might overestimate the amount of sublimation. The discrepancy in the lowest 300m of the atmosphere is not considered in this study but needs to be accounted for.*

**Page 13, 27th line. When the authors mention that "CloudSat is not able to capture individual snowfall events adequately at a single location", I think the authors should be more specific about that assertion. Indeed for specific precipitation cases, when the satellite overpasses a station closely, if the ground-radar and the CloudSat radar are properly calibrated and their Ze-Sr relations well-established, they should capture a similar precipitation rate.**

This is a correct remark by the reviewer. As both the MRRs and CloudSat apply the same detection principle, are well-calibrated and have well-established Ze-SR relations, both instruments are expected to record similar snowfall rates when operating over the exact same area. This was recently shown to be the case for a number of exact overpasses between CloudSat and the MRRs at the PE and DDU station (presentation Florentin Lemonnier at POLAR2018 conference in Davos: `Wed_8_AC-2_746`: Comparison Between Cloudsat and In-situ Radar Snowfall Rates in East Antarctica). In this work we showed that individual snowfall events cannot be captured by CloudSat when averaging over a spatial domain (i.e. a grid of 1° latitude by 2° longitude). This does not apply to very close overpasses as noted by the reviewer and has been clarified throughout the text.

In the abstract there is referred to the CloudSat product (gridded): *Moreover, the CloudSat product does not perform well in simulating individual snowfall events.*

Introduction: *Furthermore, an overview of the discrepancies between the CloudSat product and the MRR snowfall rates are identified by comparing individual snowfall events (Sect. 3.2).*

Material and methods: *Furthermore, the performance of individual event detection of the CloudSat product and ERA-Interim reanalysis is investigated.*

Results and discussion: *One must understand that the accurate total snowfall amounts obtained by CloudSat can not be attributed to the fact that the satellite is recording correct individual snowfall quantities for each grid box, but to the fact that omission and commission errors cancel each other out. Consequently, it can be concluded that the gridded CloudSat product is not the right tool to investigate individual snowfall events / synoptic events at a single location.*

Results and discussion: *As the CloudSat domain spans several tens of kilometers at a resolution of 1° latitude by 2° longitude, it often detects small snowfall events near the station. The detection of these small-scale snowfall events is the main contributor to commission errors compared to the MRRs at this spatial resolution (Fig. 6). In addition, the direct comparison between individual events detected by the MRRs and CloudSat shows a large spread and low correlation (Fig. 7). This indicates again that the gridded CloudSat product is not able to capture individual snowfall events adequately at a single location.*

Results and discussion: *For the validation and identification of individual snowfall events,*

*the ERA-Interim reanalysis product however outperforms the CloudSat-derived product.*

Conclusions: *However, for individual snowfall event identification, ERA-Interim reanalysis outperforms the gridded CloudSat product for all stations.*

Conclusions: *Apart from that, the gridded CloudSat product is not advised for the validation of individual snowfall events.*

**Page 14, 1st line. ERA-Interim reanalysis provides surface snowfall. Is it relevant to compare this surface product with 1200m a.g.l and 300m a.g.l observations? Do you take into account the effects of the low level sublimation processes on the first bin CloudSat and the first bin MRR measurements?**

As noted in the previous comments, both the CloudSat snowfall climatology achieved at 1200m a.g.l. and the observations from the MRR at 300m a.g.l. do not represent the surface snowfall amount. The goal of the paper is to evaluate the CloudSat snowfall product as an estimator of ground-based precipitation. As such it is necessary to compare with products that provide surface snowfall rates (as ERA-Interim).

In the comparison with the MRR, one needs to take into account the overestimation of snowfall amounts that is obtained from measuring at the 300m a.g.l. level. compared to the surface, which accounts for 14 %, 9 % and 7 % for respectively the PE, DDU and MZ station. In the manuscript it is clarified to take into account this discrepancy between the 300m a.g.l. level and the surface and to clarify that the goal is to evaluate the performance of CloudSat for ground-based precipitation amounts.

*An assessment of the accuracy of CloudSat as a surface snowfall product compared to ERA-Interim reanalysis is therefore viable.*

*Regarding ERA-Interim reanalysis, for both the PE and MZ station, the daily average snowfall amount is underestimated (respectively by 18 % and 45 %), while for the DDU station, ERA-Interim reanalysis outperforms the CloudSat snowfall estimate (bias is limited to 6 %). Here, one must take into account that the MRR measurements slightly overestimate the surface snowfall product (see Sect. 2.3).*

**References**

Chen, S., Hong, Y., Kulie, M., Behrangi, A., Stepanian, P. M., Cao, Q., You, Y., Zhang, J., Hu, J., and Zhang, X.: Comparison of snowfall estimates from the NASA CloudSat Cloud Profiling Radar and NOAA/NSSL Multi-Radar Multi-Sensor System, Journal of Hydrology, 541, 862–872, https://doi.org/10.1016/j.jhydrol.2016.07.047, 2016.

Grazioli, J., Madeleine, J.-B., Gallée, H., Forbes, R. M., Genthon, C., Krinner, G., and Berne, A.: Katabatic winds diminish precipitation contribution to the Antarctic ice mass balance, Proceedings of the National Academy of Sciences, 114, 10 858–10 863, https://doi.org/10.1073/pnas.1707633114, 2017.

Maahn, M., Burgard, C., Crewell, S., Gorodetskaya, I. V., Kneifel, S., Lhermitte, S., Van Tricht, K., and van Lipzig, N. P. M.: How does the spaceborne radar blind zone affect derived

surface snowfall statistics in polar regions?, Journal of Geophysical Research: Atmospheres, 119, 13 604–13 620, https://doi.org/10.1002/2014JD022079, 2014.

Palerme, C., Kay, J. E., Genthon, C., L'Ecuyer, T., Wood, N. B., and Claud, C.: How much snow falls on the Antarctic ice sheet?, The Cryosphere, 8, 1577–1587, https://doi.org/10.5194/tc-8-1577-2014, 2014.

Souverijns, N., Gossart, A., Lhermitte, S., Gorodetskaya, I. V., Kneifel, S., Maahn, M., Bliven, F. L., and van Lipzig, N. P. M.: Estimating radar reflectivity - Snowfall rate relationships and their uncertainties over Antarctica by combining disdrometer and radar observations, Atmospheric Research, 196, 211–223, https://doi.org/10.1016/j.atmosres.2017.06.001, 2017.

Wood, N. B.: Estimation of snow microphysical properties with application to millimeter-wavelength radar retrievals for snowfall rate, Ph.D. thesis, Colorado State University, Colorado, URL http://hdl.handle.net/10217/48170, 2011.

---

## Referee Report (RR1)

Florentin Lemonnier
Laboratoire de Météorologie Dynamique
Couloir 45-55, 3ème étage
75005 Paris, FRANCE

Review of the research article tc-2018-111

Dear Dr. Philip Marsh,

I have reviewed the revision of the manuscript "Evaluation of the CloudSat surface snowfall product over Antarctica using ground-based precipitation radars" by Niels Souverijns and colleagues, submitted for publication in The Cryosphere.

The authors responded pertinently to the questions I had asked. The comparison between MRR datasets and CloudSat snowfall retrieval is more relevant. However, I still have some questions before accepting this article for a publication in The Cryosphere.

Sincerely,

Florentin Lemonnier

**Science questions**

- Page 14, $3^{rd}$ line. It is mentioned that MRR is recording a significant less amount of snowfall rate than CloudSat and this by quoting Chen et al., 2016, who propose a comparative study between CloudSat CPR and NOAA/NSSL Multi-Radar Multi-Sensor System. CloudSat is known to overestimate small precipitation and underestimate large precipitation in comparison with other instruments. Is it possible that small amount of precipitation may be missed by MRRs rather than being overestimated by CloudSat, according to CloudSat CPR threshold that would allow to detect rates below 0.1 mm/hr?

---

## Author Response (AR2)

**Response to final Reviewer Comments:**
**Evaluation of the CloudSat surface snowfall product over Antarctica using ground-based precipitation radars**

Niels Souverijns, Alexandra Gossart, Stef Lhermitte, Irina V. Gorodetskaya,
Jacopo Grazioli, Claudio Duran-Alarcon, Brice Boudevillain,
Christophe Genthon, Claudio Scarchilli and Nicole P.M. van Lipzig

November 21, 2018

For clarifying our answers to the reviewers' comments, the following color scheme is used: comments of the reviewer are denoted in blue, our answers are denoted in black and quotes from the revised text are in green.

*Page 14, 3rd line. It is mentioned that MRR is recording a significant less amount of snowfall rate than CloudSat and this by quoting Chen et al., 2016, who propose a comparative study between CloudSat CPR and NOAA/NSSL Multi-Radar Multi-Sensor System. CloudSat is known to overestimate small precipitation and underestimate large precipitation in comparison with other instruments. Is it possible that small amount of precipitation may be missed by MRRs rather than being overestimated by CloudSat, according to CloudSat CPR threshold that would allow to detect rates below 0.1 mm/hr?*

The lower sensitivity bound of the MRR equals -14 dBz. Using the radar reflectivity-snowfall rate relationship of Souverijns et al. (2017), this equals to 0.004 mm/hr. For CloudSat, the threshold is even lower, so events with an intensity below this threshold can be missed. An extra sentence was added to the text.

[revised manuscript text omitted]